# FULLY-INDUCTIVE NODE CLASSIFICATION ON ARBITRARY GRAPHS

**Jianan Zhao[1,2,∗], Zhaocheng Zhu[1,2,∗], Mikhail Galkin[3,†], Hesham Mostafa[4]**
**Michael Bronstein[5,6], Jian Tang[1,7,8]**
[1]Mila - Québec AI Institute, [2]Université de Montréal, [3]Google Research [4]Intel Labs
[5]University of Oxford, [6]AITHYRA, [7]HEC Montréal, [8]CIFAR AI Chair

## ABSTRACT

One fundamental challenge in graph machine learning is generalizing to new graphs. Many existing methods following the inductive setup can generalize to test graphs with new structures, but assuming the feature and label spaces remain the same as the training ones. This paper introduces a fully-inductive setup, where models should perform inference on *arbitrary test graphs with new structures, feature and label spaces*. We propose GraphAny as the first attempt at this challenging setup. GraphAny models inference on a new graph as an analytical solution to a LinearGNN, which can be naturally applied to graphs with any feature and label spaces. To further build a stronger model with learning capacity, we fuse multiple LinearGNN predictions with learned inductive attention scores. Specifically, the attention module is carefully parameterized as a function of the entropy-normalized distance features between pairs of LinearGNN predictions to ensure generalization to new graphs. Empirically, GraphAny trained on a single Wisconsin dataset with only 120 labeled nodes can generalize to 30 new graphs with an average accuracy of 67.26%, surpassing not only all inductive baselines, but also strong transductive methods trained separately on each of the 30 test graphs.

## 1 INTRODUCTION

One of the most important requirements for machine learning models is the ability to generalize to new data. Models with better generalization abilities are able to perform better on unseen data and tasks, which is a key property for foundation models (Achiam et al., 2023; Team et al., 2023; Touvron et al., 2023) that are designed to accomplish a wide range of downstream tasks. For graphs, generalization is challenging since different graphs usually have different structures and associated attributes. Ideally, graph machine learning models are expected to accommodate this difference and learn functions that are applicable to all graphs.

Many previous works on graphs (Hamilton et al., 2017; Hang et al., 2021; Qu et al., 2022; Jang et al., 2023) consider generalization in the inductive setup, where models are supposed to perform inference on test graphs with new structures different from the training ones. However, these works rely on the assumption that the training and test graphs share the same feature and label spaces, which limits their applications to graphs in a fixed domain (e.g. social networks, citation networks). Ideally, we would like to have a model that generalizes to arbitrary graphs, involving new structures, new dimensions

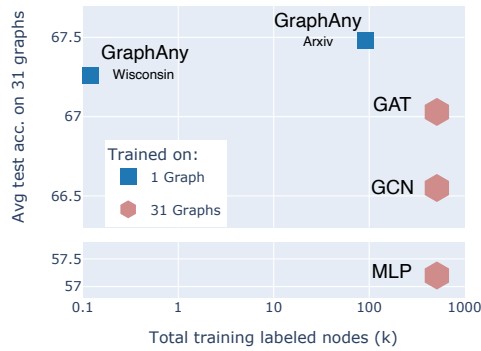

Figure 1: Average performance on 31 datasets. GraphAny is trained on a single dataset (Wisconsin or Arxiv) and performs inductive inference on any graph. The other methods have to be trained on each dataset.

∗Equal contribution. Code release: `https://github.com/DeepGraphLearning/GraphAny`
†Work done while at Intel Labs

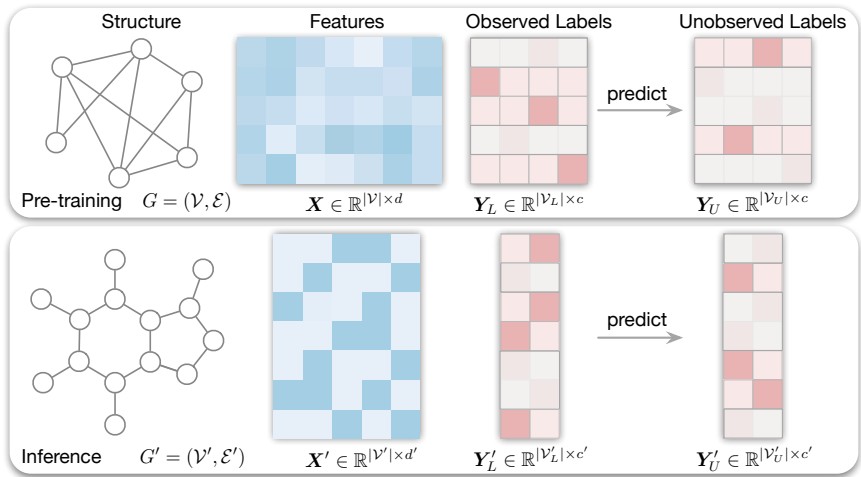

Figure 2: Fully-inductive node classification: Trained on a graph $G$, a fully-inductive model should generalize to any new graph $G'$ with new feature and label spaces ***without additional training***.

and semantics for their feature and label spaces without additional training. We name this more general and practical setting as the *fully-inductive* setup (visualized in Figure 2).

The fully-inductive setup is particularly challenging for existing graph machine learning models for two reasons: (1) Existing models learn transformations specific to the dimension, type, and structure of the features and labels used in the training, and cannot perform inference on feature and label spaces that are different from the training ones. This requires us to develop a new model architecture for arbitrary feature and label spaces. (2) Existing models learn functions specific to the training graph and cannot generalize to new graphs. This calls for an inductive function that can generalize to any graph once it is trained. Despite these challenges, it is always possible to cheat the fully-inductive setup by training a separate instance of existing models for each test dataset. We emphasize that this solution does not solve the fully-inductive setup. However, this cheating solution can be regarded as a strong baseline for fully-inductive models, since it additionally leverages back propagation on the labeled nodes and hyperparameter tuning on the validation data of the test datasets.

**Our Contributions.** We propose GraphAny to solve node classification in the fully-inductive setup. GraphAny consists of two components: *LinearGNNs* that perform inference on new feature and label spaces without training steps, and an *inductive attention* module based on entropy-normalized distance features that ensure generalization to new graphs. Specifically, our LinearGNN models the mapping between node features and labels as a non-parameteric graph convolution followed by a linear layer, whose parameters are determined in analytical form without requiring explicit training steps. While a single LinearGNN model may be far from optimal for many graphs, we employ multiple LinearGNN models with different graph convolution operators and learn an attention vector to adaptively fuse their predictions. The attention vector is carefully parameterized as a function of *distance features* between the predictions of LinearGNNs, which guarantees the model to be invariant to the permutations of feature and label dimensions. To further improve the generalization ability of our model, we propose entropy normalization to rectify the distance feature distribution to a fixed entropy, which reduces the effect of different label dimensions. By combining both modules, GraphAny learns to combine the LinearGNNs' predictions for each node based on their prediction distributions, which reflects statistics of its local structure, and generalizes to new graphs.

To summarize, the contribution of GraphAny are four folds:

- We introduce the fully-inductive setup for generalization across arbitrary graphs. Fully-inductive setup is more general and practical than the conventional inductive setup, allowing knowledge transfer across diverse domains, such as from knowledge graphs to e-commerce graphs.

- We devise LinearGNN, an analytical solution for node classification that enables efficient inductive inference on any new graph without the need for gradient descent.

- We identify two necessary properties for fully-inductive generalization: permutation invariance and dimensional robustness w.r.t. the feature and label spaces. Towards these goals, we design an inductive attention module that satisfies these properties and generalizes to new graphs.

- By combining LinearGNN and the inductive attention module, we present GraphAny, the first fully-inductive model for node classification. Across 31 datasets, GraphAny outperforms strong transductive baselines trained separately on each test dataset and achieves a $2.95\times$ speedup.

## 2 RELATED WORK

**Inductive Node Classification.** Depending on the test graphs which models generalize to, tasks on graphs can be categorized into *transductive* and *inductive* setups. In the transductive (semi-supervised) node classification setup (Kipf and Welling, 2017), test nodes belong to the same training graph the model was trained on. In the inductive setup (Hamilton et al., 2017), the test graph might be different. The majority of GNNs aiming to work in the inductive setup use known node labels as features. Hang et al. (2021) introduced *Collective Learning* GNNs with iterative prediction refinement. Jang et al. (2023) applied a diffusion model for such prediction refinement. Structured Proxy Networks (Qu et al., 2022) combined GNNs and conditional random fields (CRF) in the *structured prediction* framework. However, the train-test setup in all those works is limited to different partitions of the same bigger graph, that is, they cannot generalize to unseen graphs with different input feature dimensions and number of classes. To the best of our knowledge, GraphAny is the first approach for fully-inductive node classification, which generalizes to unseen graphs with arbitrary feature and label spaces.

**Labels as Features.** Addanki et al. (2021) demonstrated that node labels used as features yield noticeable improvements in the transductive node classification setup on MAG-240M in OGB-LSC (Hu et al., 2021a). Sato (2024) used labels as features for training-free transductive node classification with a specific GNN weight initialization strategy mimicking label propagation (hence only applicable to homophilic graphs). In homophilic graphs, node label largely depends on the labels of its close neighbors whereas in heterophilic graphs, node label does not depend on the neighboring nodes. GraphAny supports both homophilic and heterophilic graphs in both transductive and inductive setups.

**Connections to Graph Foundation Models.** Graph foundation models (GFMs), which aim to develop a single model transferable to new graphs and diverse graph tasks, have garnered significant attention in the graph learning community. The core challenge in designing a GFM is achieving *generalization across graphs with varying input and output spaces*, which requires identifying an invariant feature space that transfers effectively across graphs (Mao et al., 2024). While fine-tuning a pre-trained model on a new graph is feasible (Zhu et al., 2024), we focus on the more challenging fully-inductive setting, where the model must generalize to unseen graphs *without* additional training.

In scenarios where *graphs share a common feature space*, such as in molecular learning (Kläser et al., 2024; Kovács et al., 2023; Zhang et al., 2023; Sypetkowski et al., 2024; Shoghi et al., 2024) and material design (Batatia et al., 2024), GNNs can be applied to the shared feature space of atoms, acting as GFMs. For *non-featurized graphs*, GNNs equipped with labeling tricks (Zhang et al., 2021) have shown promise as generalizable link predictors in homogeneous graphs (Dong et al., 2024) and multi-relational knowledge graphs (Galkin et al., 2024). Another line of research (Zhao et al., 2023; Tang et al., 2023; Chai et al., 2023; Fatemi et al., 2024; Perozzi et al., 2024; Liu et al., 2024) focuses on *text-attributed graphs* (Yan et al., 2023), where features are represented as, or converted into, text. Here, large language models (LLMs) serve as universal featurizers by verbalizing the (sub)graph structure and the associated task in natural language. However, it is unclear whether the sequential nature of LLMs is suitable for graphs with permutation invariance.

In summary, while existing GFMs have shown promising results, their generalization capabilities often rely on strong assumptions over the training and test graphs, particularly the presence of a shared input space. In contrast, GraphAny is the first attempt to the more general and challenging problem of generalizing across *arbitrary graphs*. This broader scope opens up new possibilities for transferring knowledge between different graph domains, such as from knowledge graphs to e-commerce graphs. Additionally, we believe our proposed model, along with essential generalization properties like permutation invariance and dimensional robustness, provide a solid foundation for future research on powerful and versatile GFMs.

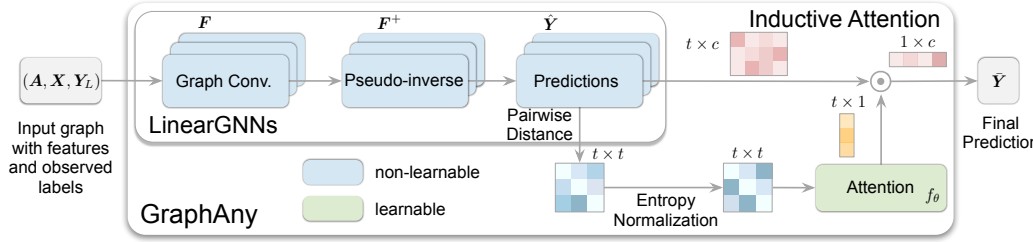

Figure 3: Overview of GraphAny: LinearGNNs are used to perform non-parametric predictions and derives the entropy-normalized distance features. The final prediction is generated by fusing multiple LinearGNN predictions on each node with an attention learned based on the distance features.

## 3 GRAPHANY: FULLY-INDUCTIVE NODE CLASSIFICATION ON ANY GRAPH

Our goal is to devise a fully-inductive model that can perform inductive inference on any new graph with arbitrary feature and label spaces, typically different from the ones associated with the training graph (Figure 2). Here we propose such a solution GraphAny, which consists of two main components (Figure 3): a LinearGNN and an attention module. Each LinearGNN provides a basic solution to inductive inference on new graphs with arbitrary feature and label spaces, while the attention module learns to combine multiple LinearGNNs based on inductive features that generalize to new graphs.

Formally, in a semi-supervised node classification task, we are given a graph $\mathcal{G} = (\mathcal{V}, \mathcal{E})$, typically represented as an adjacency matrix $\boldsymbol{A}$, and node features $\boldsymbol{X} \in \mathbb{R}^{|\mathcal{V}| \times d}$, a set of labeled nodes $\mathcal{V}_L$ and their labels $\boldsymbol{Y}_L \in \mathbb{R}^{|\mathcal{V}_L| \times c}$, where $c$ is the number of unique label classes. The goal of node classification is to predict the labels $\hat{\boldsymbol{Y}}$ for all the unlabeled nodes $\mathcal{V}_U = \mathcal{V} \setminus \mathcal{V}_L$ in the graph. In the conventional transductive learning setup, this is performed by training a GNN (e.g. GCN (Kipf and Welling, 2017), GAT (Velickovic et al., 2018)) on the subset of labeled nodes using standard backpropagation requiring multiple gradient steps. Such a GNN assumes the full graph to be given and typically does not generalize to a new graph out of the box without some forms of re-training or fine-tuning. Conversely, a fully-inductive model is expected to predict the labels $\hat{\boldsymbol{Y}}$ for any graph without expensive gradient steps. Furthermore, when a new graph is provided, it might have different dimensionality $d'$ of the features and number of class labels, $c'$, which can also appear in an arbitrary permuted order.

### 3.1 INDUCTIVE INFERENCE WITH LINEARGNNS

A key idea of this paper is to use simple GNN models whose parameters can be expressed analytically. Following existing works that simplifies GNN (Wu et al., 2019; Zhu and Koniusz, 2021; Yoo et al., 2023) by removing non-linearity, we leverage graph convolutions to process node features, followed by a linear layer to predict the labels,

$$\hat{\boldsymbol{Y}} = \mathrm{softmax}(\boldsymbol{F}\boldsymbol{W}), \tag{1}$$

where $\boldsymbol{F} = \boldsymbol{A}^k\boldsymbol{X} \in \mathbb{R}^{|\mathcal{V}| \times d}$ is the processed features and $\boldsymbol{W} \in \mathbb{R}^{d \times c}$ is the weight of the linear layer. Originally, the parameters of most existing GNNs are trained to minimize a cross-entropy loss on node classification, which does not have analytical solution and requires gradient descent to learn the weights. Alternatively, we propose to use a mean-squared error loss for optimizing the weights:

$$\boldsymbol{W}^* = \arg\min_{\boldsymbol{W}} \|\hat{\boldsymbol{Y}}_L - \boldsymbol{Y}_L\|^2, \tag{2}$$

where we use $\hat{\boldsymbol{Y}}_L$ to denote model predictions on the set of labeled nodes. The benefit of this approximation is that now we have an analytical solution for the optimal weights $\boldsymbol{W}^*$:

$$\boldsymbol{W}^* = \boldsymbol{F}_L^+\boldsymbol{Y}_L, \tag{3}$$

where $\boldsymbol{F}_L^+$ is the pseudo inverse of $\boldsymbol{F}_L$, and the model prediction is given by:

$$\hat{\boldsymbol{Y}} = \boldsymbol{F}\boldsymbol{F}_L^+\boldsymbol{Y}_L. \tag{4}$$

We term this architecture *LinearGNN*, as it approximates the prediction of linear GNNs (like SGC (Wu et al., 2019)) with a single forward pass. Its main advantage is that it requires no training, which makes it significantly more computationally efficient (see Section 3.3). Note that the derivation of LinearGNNs is independent of the graph convolution operation, and can be applied to other linear convolution operations as well. We stress that while we do not expect LinearGNNs to outperform existing transductive models on node classification, they provide a simple basic module for inductive inference. In Section 3.2, we will see how to combine multiple LinearGNNs to create a stronger model with inductive attention that generalizes to new graphs.

## 3.2 LEARNING INDUCTIVE ATTENTION OVER LINEARGNN PREDICTIONS

Although LinearGNNs provide basic solutions to inductive inference on new graphs, the learned weights are still *transductive*, i.e. specific to the training graph, and do not capture transferable knowledge that can be applied to unseen graphs. Besides, our experiments (Figure 7) suggest that different graphs may require LinearGNNs with convolution operations. Hence, a natural way to incorporate fully-inductive learning is to add an *inductive attention module* over a set of multiple LinearGNNs. Let $\alpha_u^{(1)}, \alpha_u^{(2)}, ..., \alpha_u^{(t)}$ denote the node-level attention over $t$ LinearGNNs. We generate the final prediction as a combination of all LinearGNN predictions:

$$\bar{\boldsymbol{y}}_u = \sum_{i=1}^{t} \alpha_u^{(i)} \hat{\boldsymbol{y}}_u^{(i)}. \tag{5}$$

While there are various ways to parameterize this attention module, finding an inductive solution is non-trivial since neural networks can easily fit the information specific to the training graph. We notice a necessary property for fully-inductive functions is that it should be robust to transformations on features and labels, such as permutation or masking on some dimensions (Figure 4). This requires our attention module, to be *permutation invariant* and *robust to dimension changes*, which motivates our design of distance features and entropy normalization respectively.

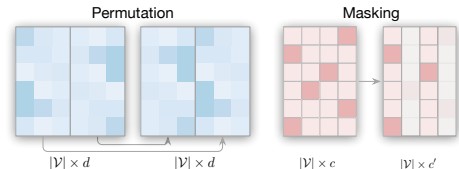

Figure 4: Transformations on graph features and labels: permutation (left), masking (right).

**Permutation-Invariant Attention with Distance Features.** We would like to design an attention module that is permutation invariant (Bronstein et al., 2021) along the data dimension.[1] Consider a new graph generated by permuting feature and label dimensions of the training graph. We expect our attention output to be invariant to these permutations in order to generate the same prediction as for the unpermutated training set.

Our idea is to construct a set of permutation-invariant features, such that any attention module we build on top of these features becomes permutation-invariant. Formally, if the permutation matrices for feature and label dimensions are $\boldsymbol{P} \in \mathbb{R}^{d \times d}$ and $\boldsymbol{Q} \in \mathbb{R}^{c \times c}$ respectively, a function $f$ is *(data) permutation-invariant* if:

$$f(\boldsymbol{X}\boldsymbol{P}, \boldsymbol{Y}_L\boldsymbol{Q}) = f(\boldsymbol{X}, \boldsymbol{Y}_L). \tag{6}$$

In our LinearGNN, the prediction $\hat{\boldsymbol{Y}}$, as a function of the new graph, is invariant to the feature permutation and equivariant to the label permutation since it has the following analytical form:

$$\hat{\boldsymbol{Y}}(\boldsymbol{X}\boldsymbol{P}, \boldsymbol{Y}_L\boldsymbol{Q}) = \boldsymbol{F}\boldsymbol{F}_L^+\boldsymbol{Y}_L\boldsymbol{Q}. \tag{7}$$

Deriving a feature that is invariant to the label permutation requires us to cancel $\boldsymbol{Q}$ with its inverse $\boldsymbol{Q}^\top$. A straightforward solution is to use a dot product feature for predictions on each node:

$$\hat{\boldsymbol{Y}}\hat{\boldsymbol{Y}}^\top = \boldsymbol{F}\boldsymbol{F}_L^+\boldsymbol{Y}_L\boldsymbol{Y}_L^\top(\boldsymbol{F}_L^+)^\top\boldsymbol{F}^\top, \tag{8}$$

---

[1]It is important to distinguish between *domain symmetry* (in this case, permutation of the nodes of the graph) and *data symmetry* (permutation of the feature and label dimensions). Invariance to domain symmetry (node permutations) is provided by design in our LinearGNN.

which is invariant to both feature and label-permutation matrices $\boldsymbol{P}$ and $\boldsymbol{Q}$. Generally, any feature that is a linear combination of dot products between LinearGNN predictions is also permutation invariant, e.g. Euclidean distance or Jensen-Shannon divergence. For a set of $t$ LinearGNN predictions $\hat{\boldsymbol{y}}_u^{(1)}, \hat{\boldsymbol{y}}_u^{(2)}, \cdots, \hat{\boldsymbol{y}}_u^{(t)}$ on a single node $u$, we construct the following $t(t-1)$ permutation-invariant features to capture their squared distances:

$$\|\hat{\boldsymbol{y}}_u^{(1)} - \hat{\boldsymbol{y}}_u^{(2)}\|^2, \|\hat{\boldsymbol{y}}_u^{(1)} - \hat{\boldsymbol{y}}_u^{(3)}\|^2, ..., \|\hat{\boldsymbol{y}}_u^{(t)} - \hat{\boldsymbol{y}}_u^{(t-1)}\|^2. \tag{9}$$

We include detailed proofs in Appendix A. An advantage of such permutation-invariant features is that any model taking them as input is also permutation invariant, allowing us to use a simple model, such as multi-layer perceptrons (MLP), to predict the attention scores over different LinearGNNs.

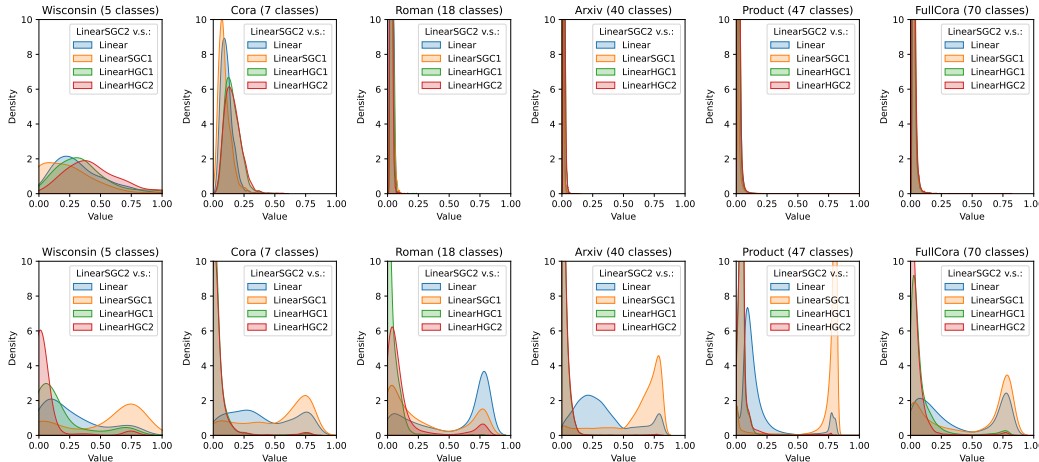

Figure 5: Comparison of Euclidean distances (the first row) and entropy-normalized (the second row) features between five channels: $\boldsymbol{F} = \boldsymbol{X}$ (Linear), $\boldsymbol{F} = \bar{\boldsymbol{A}}\boldsymbol{X}$ (LinearSGC1), $\boldsymbol{F} = \bar{\boldsymbol{A}}^2\boldsymbol{X}$ (LinearSGC2), $\boldsymbol{F} = (\boldsymbol{I} - \bar{\boldsymbol{A}})\boldsymbol{X}$ (LinearHGC1) and $\boldsymbol{F} = (\boldsymbol{I} - \bar{\boldsymbol{A}})^2\boldsymbol{X}$ (LinearHGC2) with $\bar{\boldsymbol{A}}$ denoting the row normalized adjaceny matrix. Entropy-normalized features are on the same scale and exhibit transferrable patterns across datasets.

**Robust Dimension Generalization with Entropy Normalization.** While the distance features for inductive attention ensure a permutation-invariant attention module, the distance features are known to suffer from the curse of dimensionality, where distances between vectors with larger dimensions have smaller scales (Beyer et al., 1999). This will hamper the generalization performance when the dimensions of label spaces vary across training and inference graphs (e.g. training on 7 classes for Cora and inference on 70 classes for FullCora). Empirically, as shown in the first row of Figure 5, the scale of Euclidean distance distributions decreases drastically when the number of classes increases, hampering the generalization ability of the attention module trained on a single graph.

A naive approach is to normalize the distance-feature distributions using a hyperparameter (e.g., temperature). However, due to the varying neighbor patterns across graphs (or even nodes (Luan et al., 2022)), a single hyperparameter may not be suitable for all nodes and graphs. Instead, we propose an adaptive solution that normalizes distance features to a consistent scale. To achieve this, we employ *entropy normalization*, a technique commonly used in manifold learning (Hinton and Roweis, 2002; van der Maaten and Hinton, 2008) to adaptively determine the similarity features. For node $u$, the asymmetric similarity feature between LinearGNN predictions $i$ and $j$ is defined as:

$$p_u(j|i) = \frac{\exp(-\|\hat{\boldsymbol{y}}_u^{(i)} - \hat{\boldsymbol{y}}_u^{(j)}\|^2/2(\sigma_u^{(i)})^2)}{\sum_{k \neq i} \exp(-\|\hat{\boldsymbol{y}}_u^{(i)} - \hat{\boldsymbol{y}}_u^{(k)}\|^2/(\sigma_u^{(i)})^2)}, \tag{10}$$

where $\sigma_u^{(i)}$ is the standard deviation of an isotropic multivariate Gaussian, determined by matching the entropy of distance distributions $P_u^{(i)} = \{p_u(j \mid i) \mid j \in [1, t]\}$ to a fixed hyperparameter $H$. Since the similarity features are derived from distance features, they are also permutation-invariant to the feature and label dimensions of the graph. Intuitively, this imposes a soft constraint on the

number of LinearGNN predictions considered similar to $\hat{\boldsymbol{y}}_u^{(i)}$, significantly reducing the gap between training and test features. As shown in the second row of Figure 5, entropy-normalized feature distributions are on consistent scales across datasets. Additionally, we observe that different types of homophilic graphs (e.g., the citation graph Cora and the e-commerce graph Product) exhibit similar entropy-normed features. We will further verify the effectiveness of entropy normalization empirically in Section 4.4.

### 3.3 EFFICIENT TRAINING AND FULLY-INDUCTIVE INFERENCE

In this section, we summarize how GraphAny utilizes the techniques introduced in the previous section and derive an efficient fully-inductive node classification model. As shown in Figure 3, given any graph, GraphAny first utilizes $t$ LinearGNNs to provide basic predictions with different channels. Then, entropy-normalized features are computed based on the distances between these predictions, leading to $t(t-1)$-dimensional features. Further, an inductive attention module $f_\theta$ (e.g. MLP) is used to compute the attention scores for fusing different predictions into a final prediction (Eq. 5). Since the only trainable module of GraphAny lies in the inductive attention module $f_\theta : \mathbb{R}^{t(t-1)} \to \mathbb{R}^t$, which is *independent* to feature dimension $d$ and label dimension $c$, GraphAny enjoys fully-inductive inference on any graph with arbitrary feature and label dimensions.

One advantage of GraphAny is that it is more efficient than conventional graph neural networks (e.g. GCN (Kipf and Welling, 2017)), which is due to two reasons. First, LinearGNN leverages non-parameteric graph convolution operations, which can be preprocessed and cached for all nodes on a graph, reducing the optimization complexity to $O(|\mathcal{V}_L|)$ compared with the complexity of $O(|\mathcal{E}|)$ for standard GNNs. Second, once trained, GraphAny is ready to generalize to arbitrary graphs, eliminating the need for gradient descent on test graphs.

Table 1 shows the time complexity and total wall time of GCN, LinearGNN and GraphAny. The total wall time considers all training and inference time on 31 graphs. Even without any speed optimization in our implementation, GraphAny is 2.95× faster than the optimized DGL's (Wang et al., 2020) GCN implementation in total time. We believe the speedup can be even larger with dedicate implementations of GraphAny.

Table 1: Comparison of time complexity and wall time. Note that GCN has to be trained individually on each of the 31 graphs while GraphAny only needs 1 training graph.

| Model | Pre-processing | Optimization | Inference | Total Wall Time (31 graphs) |
|---|---|---|---|---|
| GCN | 0 | $O(|\mathcal{E}|)$ | $O(|\mathcal{E}|)$ | 18.80 min |
| LinearGNN | $O(|\mathcal{E}|)$ | $O(|\mathcal{V}_L|)$ | $O(|\mathcal{V}_U|)$ | 1.25 min (15.04×) |
| GraphAny | $O(|\mathcal{E}|)$ | $O(|\mathcal{V}_L|)$ | $O(|\mathcal{V}_U|)$ | 6.37 min (2.95×) |

## 4 EXPERIMENTS

In this section, we evaluate the performance of GraphAny against both transductive and inductive methods on 31 node classification datasets (details in Appendix B). We visualize the attention of GraphAny on different datasets, shedding light on what inductive knowledge our model has learned (Section 4.3). To provide a comprehensive understanding of the proposed techniques of GraphAny, we further conduct ablation studies on the entropy-normalized feature and attention parameterization in Section 4.4. More training and implementation details are provided in Appendix C.

### 4.1 EXPERIMENTAL SETUP

**Datasets.** We have compiled a diverse collection of 31 node classification datasets from three sources: PyG (Fey and Lenssen, 2019), DGL (Wang et al., 2020), and OGB (Hu et al., 2021b). These datasets encompass a wide range of graph types including academic collaboration networks, social networks, e-commerce networks and knowledge graphs, with sizes varying from a few hundreds to a few millions of nodes. The number of classes across these datasets ranges from 2 to 70. Detailed statistics for each dataset are provided in Appendix B.

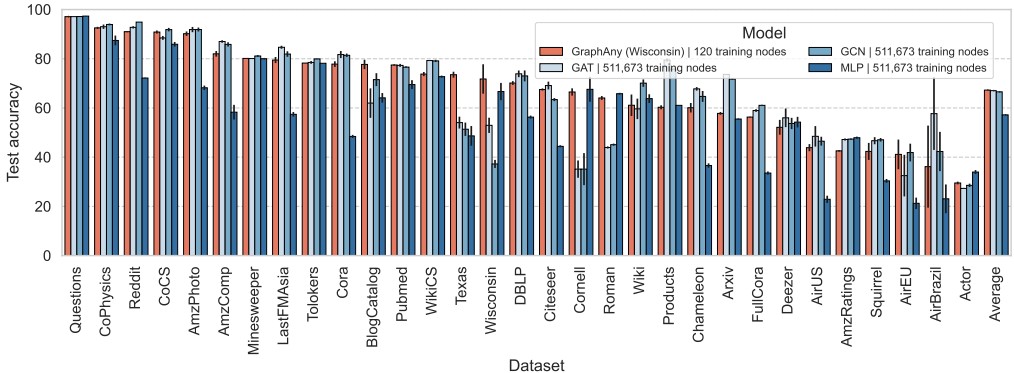

Figure 6: Inductive test accuracy (%) of GraphAny pre-trained using 120 labeled nodes of the Wisconsin dataset on 30 diverse graphs. Baseline methods are trained individually on each graph (511k labeled nodes in total). GraphAny is slightly better than the baselines in average performance.

**Implementation Details.** For GraphAny, we employ 5 LinearGNNs with different graph convolution operations: $F = X$ (Linear), $F = \bar{A}X$ (LinearSGC1), $F = \bar{A}^2 X$ (LinearSGC2), $F = (I - \bar{A})X$ (LinearHGC1) and $F = (I - \bar{A})^2 X$ (LinearHGC2) with $\bar{A}$ denoting the row normalized adjacency matrix, which cover identical, low-pass and high-pass spectral filters. In our experiments, we consider 4 GraphAny models trained separately on 4 datasets respectively: Cora (homophilic, small), Wisconsin (heterophilic, small), Arxiv (homophilic, medium), and Products (homophilic, large). The remaining 27 datasets are held out from these training sets, ensuring that the evaluations on these datasets are conducted in a fully-inductive manner. More implementation details can be found at Appendix C.

**Baselines.** As there are no existing fully-inductive node classification baselines, we include non-parametric methods (models without learnable parameters) like label propagation (Zhu and Ghahramani, 2002) and the five LinearGNNs used in GraphAny. While these methods perform inductive inference, they do not transfer knowledge across graphs. Additionally, we compare GraphAny with transductive models, including MLP, GCN (Kipf and Welling, 2017), and GAT (Velickovic et al., 2018). These models are trained separately on each dataset and serve as strong baselines for inductive models, as they benefit from backpropagation on labeled nodes and hyperparameter tuning on validation sets of the test dataset.

## 4.2 PERFORMANCE OF INDUCTIVE NODE CLASSIFICATION

Table 2 presents the results of GraphAny and various baselines on 31 node classification datasets (complete results for each dataset are provided in Appendix D). Our proposed LinearGNNs, despite being non-parametric, demonstrate competitive performance. Notably, LinearSGC2, a linear model with a two-hop graph convolution layer, achieves only 2.1% lower accuracy than GCN, which aligns with previous findings that SGC performs comparably to GCN (Wu et al., 2019). Moreover, LinearSGC2 leverages an analytical solution for inference, making it approximately $15\times$ faster than training a GCN from scratch on each dataset (see Table 1). Additionally, we observe that the optimal LinearGNN model differs across datasets, highlighting that no single graph convolution kernel is universally effective for all graphs.

As for GraphAny, which is trained on just 1 of the 31 graphs, it significantly outperforms LinearGNNs and even slightly surpasses transductive baselines that are individually trained on all 31 graphs. This improvement is primarily driven by inductive generalization, as GraphAny achieves its strongest performance on the 27 held-out (fully-inductive) datasets rather than the 4 training (transductive) datasets. A closer examination of Figure 6 shows that GraphAny performs well on both homophilic and heterophilic graphs in an inductive manner. We attribute this to the inductive attention module, which adaptively fuses predictions from different graph convolution kernels for each node. Interestingly, we also observe minimal performance differences between GraphAny when trained on small datasets (e.g., Cora and Wisconsin) and large datasets (e.g., Arxiv and Products). We hypothesize that even small datasets contain sufficiently diverse local node patterns (e.g., homophily

Table 2: Main experiment results (test accuracy %).

| Category | Method | Cora | Wisconsin | Arxiv | Products | Held Out Avg. (27 graphs) | Total Avg. (31 graphs) |
|---|---|---|---|---|---|---|---|
| **Transductive** | MLP | $48.42_{\pm0.63}$ | $66.67_{\pm3.51}$ | $55.50_{\pm0.23}$ | $61.06_{\pm0.08}$ | 57.09 | 57.20 |
| | GCN | $81.40_{\pm0.70}$ | $37.25_{\pm1.64}$ | $71.74_{\pm0.29}$ | $75.79_{\pm0.12}$ | 65.55 | 66.55 |
| | GAT | $\mathbf{81.70_{\pm1.43}}$ | $52.94_{\pm3.10}$ | $\mathbf{73.65_{\pm0.11}}$ | $\mathbf{79.45_{\pm0.59}}$ | 65.31 | 67.03 |
| **Non-parameteric** | LabelProp | $60.30_{\pm0.00}$ | $16.08_{\pm2.15}$ | $0.98_{\pm0.00}$ | $74.50_{\pm0.00}$ | $50.73_{\pm0.31}$ | $49.01_{\pm0.27}$ |
| | Linear | $52.80_{\pm0.00}$ | $\mathbf{80.00_{\pm2.15}}$ | $46.79_{\pm0.00}$ | $42.10_{\pm0.00}$ | $57.91_{\pm0.43}$ | $57.59_{\pm0.42}$ |
| | LinearSGC1 | $74.30_{\pm0.00}$ | $45.49_{\pm13.96}$ | $55.33_{\pm0.00}$ | $56.58_{\pm0.00}$ | $62.69_{\pm0.24}$ | $62.08_{\pm0.48}$ |
| | LinearSGC2 | $78.20_{\pm0.00}$ | $57.64_{\pm1.07}$ | $59.58_{\pm0.00}$ | $62.92_{\pm0.00}$ | $64.38_{\pm0.48}$ | $64.41_{\pm0.39}$ |
| | LinearHGC1 | $22.50_{\pm0.00}$ | $64.32_{\pm2.15}$ | $22.92_{\pm0.00}$ | $15.00_{\pm0.00}$ | $37.01_{\pm0.20}$ | $36.26_{\pm0.23}$ |
| | LinearHGC2 | $23.80_{\pm0.00}$ | $56.08_{\pm4.29}$ | $20.65_{\pm0.00}$ | $13.39_{\pm0.00}$ | $35.62_{\pm0.68}$ | $34.70_{\pm0.55}$ |
| **Inductive (training set)** | GraphAny (Cora) | $80.18_{\pm0.13}$ | $61.18_{\pm5.08}$ | $58.62_{\pm0.05}$ | $61.60_{\pm0.10}$ | $67.24_{\pm0.23}$ | $67.00_{\pm0.14}$ |
| | GraphAny (Wisconsin) | $77.82_{\pm1.15}$ | $71.77_{\pm5.98}$ | $57.79_{\pm0.56}$ | $60.28_{\pm0.80}$ | $67.31_{\pm0.38}$ | $67.26_{\pm0.20}$ |
| | GraphAny (Arxiv) | $79.38_{\pm0.16}$ | $65.10_{\pm3.22}$ | $58.68_{\pm0.17}$ | $61.31_{\pm0.20}$ | $67.65_{\pm0.31}$ | $67.46_{\pm0.27}$ |
| | GraphAny (Products) | $79.36_{\pm0.23}$ | $65.89_{\pm2.23}$ | $58.58_{\pm0.11}$ | $61.19_{\pm0.23}$ | $\mathbf{67.66_{\pm0.39}}$ | $\mathbf{67.48_{\pm0.33}}$ |

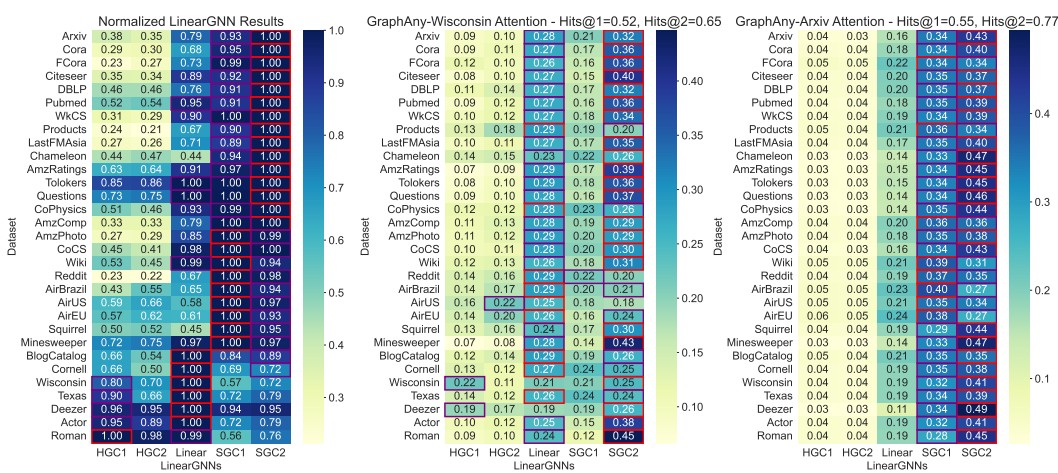

Figure 7: Normalized performance of LinearGNNs (left; best as 1.00) and attention weights of GraphAny trained on Wisconsin (middle) and Arxiv (right) respectively. The best and second best LinearGNN performance and attention weights for each dataset are highlighted with red and purple rectangles respectively. The learned inductive attention of GraphAny successfully identifies the best-performing LinearGNN for most datasets.

and heterophily) (Luan et al., 2022), enabling GraphAny to learn an effective node-level attention mechanism that generalizes well with a limited number of nodes (e.g., 120). A detailed analysis of the attention module is provided in Section 4.3.

## 4.3 VISUALIZATION OF THE INDUCTIVE ATTENTION

To understand how LinearGNNs are combined in GraphAny by the inductive attention, we visualize the attention weights of GraphAny (Wisconsin) and GraphAny (Arxiv) on all datasets, averaged across nodes. For reference, we also visualize the performance of each individual LinearGNN on all datasets. As shown in Figure 7, we can see that half of the datasets are homophilic with LinearSGC2 being the optimal LinearGNN, while the other half prefers LinearHGC1, Linear or LinearSGC1. In most cases, GraphAny successfully identifies the optimal LinearGNN within its top-2 attention weights, with Hits@2 being 0.65 and 0.77 for GraphAny trained on Wisconsin and Arxiv respectively.

We hypothesis that this amazing inductive performance comes from the inductive entropy-normed distance feature we derived, where homophilic and heterophilic graphs share different patterns (see the second row in Figure 5). Interestingly, there is a distinction between the attention distributions when training on different datasets: GraphAny-Wisconsin leads a relatively balanced distribution of attention across 5 LinearGNNs, while Graph-Arxiv prefers a more focused distribution of attention, favoring low-pass filters like LinearSGC1 and LinearSGC2. This reflects the nature of these training sets: As shown in the left part of Figure 7, all LinearGNN channels in Wisconsin are reasonably good,

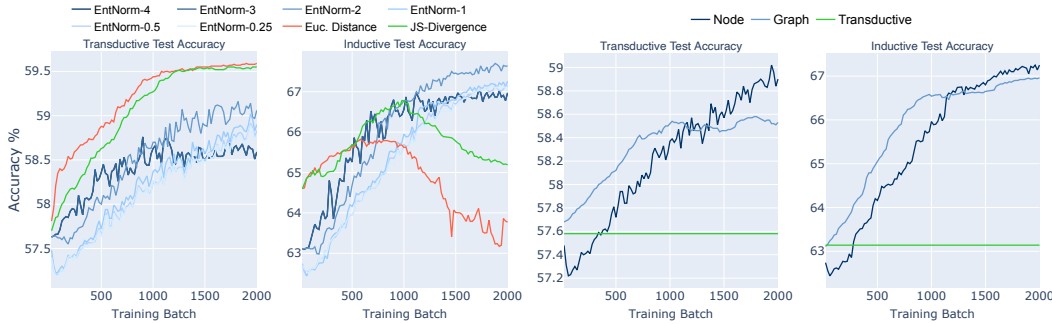

Figure 8: Performance of different distance features with and without entropy normalization.

Figure 9: Performance of different attention parameterizations.

indicating diverse message-passing pattern exists, but Arxiv is a homophilic dataset where Linear, LinearSGC1 and LinearSGC2 have better performance (Table 2). These different message-passing patterns are learned and used by GraphAny to generate inductive attention scores.

## 4.4 ABLATION STUDIES

**Entropy Normalization.** A key component of GraphAny is the entropy normalization technique applied to distance features. As discussed in Figure 5, entropy normalization ensures that the generated features are of a consistent scale. Here, we further evaluate entropy normalization from a performance perspective. Figure 8 demonstrates that unnormalized features, such as Euclidean distance and Jensen-Shannon divergence, yield better test performance in the transductive setting. However, their performance in the inductive setting decreases as training progresses, indicating that the model overfits to transductive information in the features, such as feature scale. In contrast, distance features normalized to the same entropy $H$ (denoted as EntNorm-$H$ in the figure) achieve stable convergence in both transductive and inductive settings. Additionally, entropy normalization shows robustness to the choice of the hyperparameter entropy value $H$, with different selections resulting in similar convergence rates and performance.

**Attention Parameterization.** It is known that the optimal message-passing patterns vary for different graphs (Zhu et al., 2020) or even different nodes (Luan et al., 2022). Therefore, GraphAny utilizes a *node-level attention* to adaptively combine different LinearGNN predictions. Here we consider two variants of attention parameterization: (1) *Graph-level attention* uses distance features averaged over all nodes in a training batch, which results in the same attention for all nodes in a training batch, losing the personalization for each node. (2) *Transductive attention* directly parameterizes attention weights as a $t$-dimensional vector and assumes they can transfer to new graphs. Figure 9 plots the transductive and inductive test performance curves for different attention parameterizations. We notice that transductive attention does not even learn anything useful, given its performance is worse than a single LinearSGC2 model (see Table 2). Comparing node-level attention and graph-level attention, we can see that node-level attention converges slower than graph-level attention, but results in better performance in both transductive and inductive settings. This suggests the effectiveness of learning fine-grained attention based on the local information of each node.

## 5 CONCLUSION

In this paper, we propose GraphAny, the first fully-inductive node classification model capable of performing inference on any graph with arbitrary feature or label space. GraphAny is composed of two core components: LinearGNNs and an inductive attention module. LinearGNNs enable efficient inductive inference on unseen graphs, while the inductive attention module learns to adaptively aggregate predictions from multiple LinearGNNs. Trained on a single graph, GraphAny demonstrates strong generalization to 30 new graphs, even surpassing the average performance of transductive models that are trained separately on each dataset.

ACKNOLEDGEMENT

The authors would like to thank Dinghuai Zhang for his helpful discussions and comments. This project is supported by the Intel-MILA partnership program, the Natural Sciences and Engineering Research Council (NSERC) Discovery Grant, the Canada CIFAR AI Chair Program, collaboration grants between Microsoft Research and Mila, Tencent AI Lab Rhino-Bird Gift Fund and a NRC Collaborative R&D Project (AI4DCORE-06). This project was also partially funded by IVADO Fundamental Research Project grant PRF-2019-3583139727. M.B. is partially supported by the EPSRC Turing AI World-Leading Research Fellowship No. EP/X040062/1 and EPSRC AI Hub No. EP/Y028872/1. The computation resource of this project is supported by Mila, Calcul Québec and the Digital Research Alliance of Canada.

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

## A  PROOF OF FEATURE AND LABEL PERMUTATION INVARIANCE

In this section, we prove that the proposed distance-based features have the desiring property of feature- and label-permutation invariance.

### A.1  LABEL-PERMUTATION INVARIANCE

Here, we show the label-permutation invariance of the distance of LinearGNN predictions. I.e. for any two LinearGNNs $i$ and $j$, the (squared) distances between the permuted predictions, denoted as $\|\tilde{\boldsymbol{y}}_u^{(i)} - \tilde{\boldsymbol{y}}_u^{(j)})\|^2$ and the distances between the original predictions, i.e. $\|\hat{\boldsymbol{y}}_u^{(i)} - \hat{\boldsymbol{y}}_u^{(j)}\|^2$ are the same.

Consider permuting the label dimension, i.e. right multiplicate a permutation matrix $\boldsymbol{Q} \in \mathbb{R}^{c \times c}$ to the LinearGNN prediction $\hat{\boldsymbol{Y}} = \boldsymbol{F}\boldsymbol{F}_L^+ \boldsymbol{Y}_L$. (Eq 4). The permutated label logits $\tilde{\boldsymbol{Y}}$ can be expressed as:

$$\tilde{\boldsymbol{Y}} = \boldsymbol{F}\boldsymbol{F}_L^+ \boldsymbol{Y}_L \boldsymbol{Q}. \tag{11}$$

Given the linearity of matrix multiplication, we have:

$$\tilde{\boldsymbol{Y}} = \boldsymbol{F}\boldsymbol{F}_L^+ (\boldsymbol{Y}_L \boldsymbol{Q}) = (\boldsymbol{F}\boldsymbol{F}_L^+ \boldsymbol{Y}_L)\boldsymbol{Q} = \hat{\boldsymbol{Y}}\boldsymbol{Q}. \tag{12}$$

Thus, label-permutation *equivariance* is proved. Since the permutation matrix $\mathbf{Q}$ is orthonormal (i.e. $\mathbf{Q}^T\mathbf{Q} = \mathbf{I}$), pairwise distances between permuted logits remain the same as those between the original logits.

$$\begin{aligned}
\|\tilde{\boldsymbol{y}}_u^{(i)} - \tilde{\boldsymbol{y}}_u^{(j)}\|^2 &= \hat{\boldsymbol{y}}_u^{(i)T}\boldsymbol{Q}^T\boldsymbol{Q}\hat{\boldsymbol{y}}_u^{(i)} + \hat{\boldsymbol{y}}_u^{(j)T}\boldsymbol{Q}^T\boldsymbol{Q}\hat{\boldsymbol{y}}_u^{(j)} - 2\hat{\boldsymbol{y}}_u^{(i)T}\boldsymbol{Q}^T\boldsymbol{Q}\hat{\boldsymbol{y}}_u^{(j)} \\
&= \hat{\boldsymbol{y}}_u^{(i)T}\hat{\boldsymbol{y}}_u^{(i)} + \hat{\boldsymbol{y}}_u^{(j)T}\hat{\boldsymbol{y}}_u^{(j)} - 2\hat{\boldsymbol{y}}_u^{(i)T}\hat{\boldsymbol{y}}_u^{(j)} \\
&= \|\hat{\boldsymbol{y}}_u^{(i)} - \hat{\boldsymbol{y}}_u^{(j)}\|^2.
\end{aligned} \tag{13}$$

Hence, the distance-based feature is label permutation-invariant.

### A.2  FEATURE-PERMUTATION INVARIANCE

Now, let's consider permuting the feature dimension by right multiplication with a permutation matrix $\boldsymbol{P} \in \mathbb{R}^{d \times d}$, resulting in permuted features on labeled nodes denoted as $\boldsymbol{F}_L\boldsymbol{P}$. Since the permutation matrix $\boldsymbol{P}$ is orthonormal, the pseudoinverse of $\boldsymbol{F}_L\boldsymbol{P}$ is:

$$(\boldsymbol{F}_L\boldsymbol{P})^+ = \boldsymbol{P}^\top \boldsymbol{F}_L^+. \tag{14}$$

Substituting it into the equation 4, the predictions using permuted feature are:

$$\boldsymbol{F}\boldsymbol{P}(\boldsymbol{F}_L\boldsymbol{P})^+ \boldsymbol{Y}_L = \boldsymbol{F}\boldsymbol{P}\boldsymbol{P}^\top \boldsymbol{F}_L^+ \boldsymbol{Y}_L = \boldsymbol{F}\boldsymbol{F}_L^+ \boldsymbol{Y}_L = \hat{\boldsymbol{Y}}. \tag{15}$$

Here, $\boldsymbol{P}\boldsymbol{P}^\top = \boldsymbol{I}$ where $\boldsymbol{I} \in \mathbb{R}^{d \times d}$ is the identity matrix, hence proving that the predictions with feature-permutation are identical to the original predictions $\hat{\boldsymbol{Y}}$. That is, the predictions are feature-permutation invariant. Therefore, the distance features are also feature-permutation invariant.

## B  DATASETS

We collect a diverse collection of 31 node classification datasets from three sources: PyG (Fey and Lenssen, 2019), DGL (Wang et al., 2020), and OGB (Hu et al., 2021b). We follow the default split if there is a given one, otherwise, we use the standard semi-supervised setting (Kipf and Welling, 2017) where 20 nodes are randomly selected as training nodes for each label. The detailed dataset information is summarized in Table 3.

Table 3: 31 datasets used in this paper. Of those, 20 are homophilic and 11 are heterophilic.

| Dataset | #Nodes | #Edges | #Feature | #Classes | #Labeled Nodes | Train/Val/Test Ratios (%) | Category | Source |
|---|---|---|---|---|---|---|---|---|
| Air Brazil | 131 | 1074 | 131 | 4 | 80 | 61.1/19.1/19.8 | Homophilic | Ribeiro et al. (2017) |
| Cornell | 183 | 554 | 1703 | 5 | 87 | 47.5/32.2/20.2 | Heterophilic | Pei et al. (2020) |
| Texas | 183 | 558 | 1703 | 5 | 87 | 47.5/31.7/20.2 | Heterophilic | Pei et al. (2020) |
| Wisconsin | 251 | 900 | 1703 | 5 | 120 | 47.8/31.9/20.3 | Heterophilic | Pei et al. (2020) |
| Air EU | 399 | 5995 | 399 | 4 | 80 | 20.1/39.8/40.1 | Homophilic | Ribeiro et al. (2017) |
| Air US | 1190 | 13599 | 1190 | 4 | 80 | 6.7/46.6/46.6 | Homophilic | Ribeiro et al. (2017) |
| Chameleon | 2277 | 36101 | 2325 | 5 | 1092 | 48.0/32.0/20.0 | Heterophilic | Pei et al. (2020) |
| Wiki | 2405 | 17981 | 4973 | 17 | 340 | 14.1/42.9/43.0 | Homophilic | Yang et al. (2023) |
| Cora | 2708 | 10556 | 1433 | 7 | 140 | 5.2/18.5/36.9 | Homophilic | Yang et al. (2016) |
| Citeseer | 3327 | 9104 | 3703 | 6 | 120 | 3.6/15.0/30.1 | Homophilic | Yang et al. (2016) |
| BlogCatalog | 5196 | 343486 | 8189 | 6 | 120 | 2.3/48.8/48.8 | Homophilic | Yang et al. (2023) |
| Squirrel | 5201 | 217073 | 2089 | 5 | 2496 | 48.0/32.0/20.0 | Heterophilic | Pei et al. (2020) |
| Actor | 7600 | 30019 | 932 | 5 | 3648 | 48.0/32.0/20.0 | Heterophilic | Pei et al. (2020) |
| LastFM Asia | 7624 | 55612 | 128 | 18 | 360 | 4.7/47.6/47.6 | Homophilic | Rozemberczki et al. (2021) |
| AmzPhoto | 7650 | 238162 | 745 | 8 | 160 | 2.1/49.0/49.0 | Homophilic | Shchur et al. (2018) |
| Minesweeper | 10000 | 78804 | 7 | 2 | 5000 | 50.0/25.0/25.0 | Heterophilic | Platonov et al. (2023) |
| WikiCS | 11701 | 431206 | 300 | 10 | 580 | 5.0/15.1/49.9 | Homophilic | Mernyei and Cangea (2020) |
| Tolokers | 11758 | 1038000 | 10 | 2 | 5879 | 50.0/25.0/25.0 | Heterophilic | Platonov et al. (2023) |
| AmzComp | 13752 | 491722 | 767 | 10 | 200 | 1.5/49.3/49.3 | Homophilic | Shchur et al. (2018) |
| DBLP | 17716 | 105734 | 1639 | 4 | 80 | 0.5/49.8/49.8 | Homophilic | Bojchevski and Günnemann (2018) |
| CoCS | 18333 | 163788 | 6805 | 15 | 300 | 1.6/49.2/49.2 | Homophilic | Shchur et al. (2018) |
| Pubmed | 19717 | 88648 | 500 | 3 | 60 | 0.3/2.5/5.1 | Homophilic | Yang et al. (2016) |
| FullCora | 19793 | 126842 | 8710 | 70 | 1400 | 7.1/46.5/46.5 | Homophilic | Bojchevski and Günnemann (2018) |
| Roman Empire | 22662 | 65854 | 300 | 18 | 11331 | 50.0/25.0/25.0 | Heterophilic | Platonov et al. (2023) |
| Amazon Ratings | 24492 | 186100 | 300 | 5 | 12246 | 50.0/25.0/25.0 | Heterophilic | Platonov et al. (2023) |
| Deezer | 28281 | 185504 | 128 | 2 | 40 | 0.1/49.9/49.9 | Homophilic | Rozemberczki et al. (2021) |
| CoPhysics | 34493 | 495924 | 8415 | 5 | 100 | 0.3/49.9/49.9 | Homophilic | Shchur et al. (2018) |
| Questions | 48921 | 307080 | 301 | 2 | 24460 | 50.0/25.0/25.0 | Heterophilic | Platonov et al. (2023) |
| Arxiv | 169343 | 1166243 | 128 | 40 | 90941 | 53.7/17.6/28.7 | Homophilic | Hu et al. (2021b) |
| Reddit | 232965 | 114615892 | 602 | 41 | 153431 | 65.9/10.2/23.9 | Homophilic | Hamilton et al. (2017) |
| Product | 2449029 | 123718280 | 100 | 47 | 196615 | 8.0/1.6/90.4 | Homophilic | Hu et al. (2021b) |

## C IMPLEMENTATION DETAILS

All experiments were conducted using five different random seeds: {0, 1, 2, 3, 4}. The best hyperparameters were selected based on the validation accuracy. The runtime measurements presented in Table 1 were performed on an NVIDIA Quadro RTX 8000 GPU with CUDA version 12.2, supported by an AMD EPYC 7502 32-Core Processor that features 64 cores and a maximum clock speed of 2.5 GHz. All GraphAny experiments can be conducted on a single GPU with 20GB GPU memory and 50GB CPU memory.

### C.1 GRAPHANY IMPLEMENTATION

Table 4: Summary of hyperparameters of GraphAny on different datasets.

| Dataset | # Batches | Learning Rate | Hidden Dimension | # MLP Layers | Entropy |
|---|---|---|---|---|---|
| Cora | 500 | 0.0002 | 64 | 1 | 2 |
| Wisconsin | 1000 | 0.0002 | 32 | 2 | 1 |
| Arxiv | 1000 | 0.0002 | 128 | 2 | 1 |
| Products | 1000 | 0.0002 | 128 | 2 | 1 |

We optimize the attention module using standard cross-entropy loss on the labeled nodes. In each training batch (batch size set as 128 for all experiments), we randomly sample two disjoint sets of nodes from the labeled nodes $\mathcal{V}_L$: $\mathcal{V}_{\text{ref}}$ and $\mathcal{V}_{\text{target}}$. $\mathcal{V}_{\text{ref}}$ is used for performing inference using LinearGNNs, and $\mathcal{V}_{\text{target}}$ is utilized to compute the loss and update the attention module. This separation is intended to prevent the attention module from learning trivial solutions unless the number of labeled nodes is too low to allow for a meaningful split. Empirically, as the final attention weights of GraphAny mostly focus on Linear, LinearSGC1, and LinearSGC2 (Figure 7), we mask the attention for LinearHGC1 and LinearHGC2 to achieve faster convergence.

The hyperparameter search space for GraphAny is relatively small, we fixed the batch size as 128 and varied the number of training batches with options of 500, 1000, and 1500; explored hidden dimensions of 32, 64, and 128; tested configurations with 1, 2, and 3 MLP layers; and set the fixed entropy value $H$ at 1 and 2. The optimal settings derived from this hyperparameter search space are detailed in Table 4.

## C.2 BASELINE IMPLEMENTATION

We utilize the Deep Graph Library (DGL) (Wang et al., 2020) implementations of GCN and GAT for our baseline models. To optimize their performance, we conducted a comprehensive hyperparameter tuning using a grid search strategy on each dataset. The search space included the following parameters: number of epochs fixed at 400; hidden dimensions explored were 64, 128, and 256; number of layers tested included 1, 2, and 3; and the learning rates considered were 0.0002, 0.0005, 0.001, and 0.002.

For label propagation, we use the DGL's implementation and use grid search to find the best model with a search space defined as follows: number of propagation hops of 1, 2, and 3; $\alpha$ of 0.1, 0.3, 0.5, 0.7, and 0.9.

## D COMPLETE RESULTS

Table 5 provides all results on 31 datasets for MLP, GCN, GAT baselines trained on each dataset from scratch and four GraphAny models trained only on one graph.

Table 5: Per-dataset results of all baselines and four GraphAny models

| Dataset | MLP | GCN | GAT | GraphAny (Products) | GraphAny (Arxiv) | GraphAny (Wisconsin) | GraphAny (Cora) |
|---|---|---|---|---|---|---|---|
| Actor | $33.95_{\pm 0.80}$ | $28.55_{\pm 0.68}$ | $27.30_{\pm 0.22}$ | $28.99_{\pm 0.61}$ | $28.60_{\pm 0.21}$ | $29.51_{\pm 0.55}$ | $27.91_{\pm 0.16}$ |
| AirBrazil | $23.08_{\pm 5.83}$ | $42.31_{\pm 7.98}$ | $57.69_{\pm 14.75}$ | $34.61_{\pm 16.54}$ | $34.61_{\pm 16.09}$ | $36.15_{\pm 16.68}$ | $33.07_{\pm 16.68}$ |
| AirEU | $21.25_{\pm 2.31}$ | $41.88_{\pm 3.60}$ | $32.50_{\pm 8.45}$ | $41.75_{\pm 6.84}$ | $41.50_{\pm 6.50}$ | $41.13_{\pm 6.02}$ | $40.50_{\pm 7.01}$ |
| AirUS | $22.88_{\pm 1.46}$ | $46.49_{\pm 1.81}$ | $48.47_{\pm 4.17}$ | $43.57_{\pm 2.07}$ | $43.64_{\pm 1.83}$ | $43.86_{\pm 1.44}$ | $43.46_{\pm 1.45}$ |
| AmzComp | $58.28_{\pm 2.98}$ | $85.83_{\pm 0.86}$ | $87.01_{\pm 0.50}$ | $82.90_{\pm 1.25}$ | $83.04_{\pm 1.24}$ | $82.00_{\pm 1.14}$ | $82.99_{\pm 1.22}$ |
| AmzPhoto | $68.20_{\pm 0.88}$ | $91.88_{\pm 0.79}$ | $91.86_{\pm 1.07}$ | $90.64_{\pm 0.82}$ | $90.60_{\pm 0.82}$ | $90.18_{\pm 0.91}$ | $90.14_{\pm 0.93}$ |
| AmzRatings | $47.90_{\pm 0.45}$ | $47.35_{\pm 0.26}$ | $47.18_{\pm 0.42}$ | $42.70_{\pm 0.10}$ | $42.74_{\pm 0.12}$ | $42.57_{\pm 0.34}$ | $42.84_{\pm 0.04}$ |
| BlogCatalog | $64.11_{\pm 1.95}$ | $71.51_{\pm 2.62}$ | $61.98_{\pm 5.99}$ | $74.73_{\pm 3.19}$ | $73.63_{\pm 2.95}$ | $77.69_{\pm 1.90}$ | $72.52_{\pm 3.22}$ |
| Chameleon | $36.62_{\pm 0.87}$ | $64.69_{\pm 2.21}$ | $67.76_{\pm 0.72}$ | $62.59_{\pm 0.87}$ | $62.59_{\pm 0.86}$ | $60.09_{\pm 1.93}$ | $61.49_{\pm 1.88}$ |
| Citeseer | $44.40_{\pm 0.44}$ | $63.40_{\pm 0.63}$ | $69.10_{\pm 1.59}$ | $67.94_{\pm 0.29}$ | $68.34_{\pm 0.23}$ | $67.50_{\pm 0.44}$ | $68.90_{\pm 0.07}$ |
| CoCS | $85.88_{\pm 0.93}$ | $91.83_{\pm 0.71}$ | $88.47_{\pm 0.79}$ | $90.46_{\pm 0.54}$ | $90.45_{\pm 0.59}$ | $90.85_{\pm 0.63}$ | $90.47_{\pm 0.63}$ |
| CoPhysics | $87.43_{\pm 1.98}$ | $93.93_{\pm 0.37}$ | $93.01_{\pm 0.89}$ | $92.66_{\pm 0.52}$ | $92.69_{\pm 0.52}$ | $92.54_{\pm 0.43}$ | $92.70_{\pm 0.54}$ |
| Cora | $48.42_{\pm 0.63}$ | $81.40_{\pm 0.70}$ | $81.70_{\pm 1.43}$ | $79.36_{\pm 0.23}$ | $79.38_{\pm 0.16}$ | $77.82_{\pm 1.15}$ | $80.18_{\pm 0.13}$ |
| Cornell | $67.57_{\pm 5.06}$ | $35.14_{\pm 6.51}$ | $35.14_{\pm 3.52}$ | $64.86_{\pm 0.00}$ | $65.94_{\pm 1.48}$ | $66.49_{\pm 1.48}$ | $64.86_{\pm 1.91}$ |
| DBLP | $56.27_{\pm 0.62}$ | $73.02_{\pm 2.22}$ | $73.87_{\pm 1.35}$ | $70.62_{\pm 0.97}$ | $70.90_{\pm 0.88}$ | $70.13_{\pm 0.77}$ | $71.73_{\pm 0.94}$ |
| Deezer | $54.24_{\pm 2.15}$ | $53.69_{\pm 2.29}$ | $55.99_{\pm 3.78}$ | $52.09_{\pm 2.78}$ | $52.11_{\pm 2.79}$ | $52.13_{\pm 3.02}$ | $51.98_{\pm 2.79}$ |
| LastFMAsia | $57.41_{\pm 0.93}$ | $81.91_{\pm 1.12}$ | $84.66_{\pm 0.41}$ | $80.17_{\pm 0.44}$ | $80.60_{\pm 0.58}$ | $79.47_{\pm 1.23}$ | $80.83_{\pm 0.41}$ |
| Minesweeper | $80.00_{\pm 0.00}$ | $81.12_{\pm 0.37}$ | $80.08_{\pm 0.04}$ | $80.27_{\pm 0.16}$ | $80.30_{\pm 0.13}$ | $80.13_{\pm 0.09}$ | $80.46_{\pm 0.15}$ |
| Pubmed | $69.50_{\pm 1.79}$ | $76.60_{\pm 0.32}$ | $77.30_{\pm 0.60}$ | $76.54_{\pm 0.34}$ | $76.36_{\pm 0.17}$ | $77.46_{\pm 0.30}$ | $76.60_{\pm 0.31}$ |
| Questions | $97.33_{\pm 0.06}$ | $97.15_{\pm 0.04}$ | $97.11_{\pm 0.02}$ | $97.10_{\pm 0.01}$ | $97.09_{\pm 0.02}$ | $97.11_{\pm 0.00}$ | $97.06_{\pm 0.03}$ |
| Reddit | $72.16_{\pm 0.15}$ | $94.89_{\pm 0.02}$ | $92.76_{\pm 0.46}$ | $90.67_{\pm 0.13}$ | $90.58_{\pm 0.12}$ | $91.00_{\pm 0.24}$ | $90.46_{\pm 0.03}$ |
| Roman | $65.80_{\pm 0.35}$ | $45.08_{\pm 0.43}$ | $43.93_{\pm 0.45}$ | $64.66_{\pm 0.84}$ | $64.25_{\pm 1.09}$ | $64.06_{\pm 0.78}$ | $64.25_{\pm 0.64}$ |
| Squirrel | $30.36_{\pm 0.78}$ | $47.07_{\pm 0.71}$ | $46.69_{\pm 1.44}$ | $49.45_{\pm 0.67}$ | $49.70_{\pm 0.95}$ | $42.34_{\pm 3.46}$ | $48.49_{\pm 0.98}$ |
| Texas | $48.65_{\pm 4.01}$ | $51.35_{\pm 2.71}$ | $54.05_{\pm 2.41}$ | $73.52_{\pm 2.96}$ | $72.97_{\pm 2.71}$ | $73.51_{\pm 1.21}$ | $71.89_{\pm 1.48}$ |
| Tolokers | $78.16_{\pm 0.02}$ | $79.93_{\pm 0.10}$ | $78.50_{\pm 0.55}$ | $78.18_{\pm 0.03}$ | $78.18_{\pm 0.04}$ | $78.24_{\pm 0.03}$ | $78.20_{\pm 0.02}$ |
| Wiki | $63.79_{\pm 1.77}$ | $70.09_{\pm 1.51}$ | $59.63_{\pm 4.16}$ | $63.08_{\pm 3.61}$ | $62.96_{\pm 3.68}$ | $61.10_{\pm 4.36}$ | $60.56_{\pm 3.62}$ |
| Wisconsin | $66.67_{\pm 3.51}$ | $37.25_{\pm 1.64}$ | $52.94_{\pm 3.10}$ | $65.89_{\pm 2.23}$ | $65.10_{\pm 3.22}$ | $71.77_{\pm 5.98}$ | $61.18_{\pm 5.08}$ |
| WikiCS | $72.72_{\pm 0.43}$ | $79.12_{\pm 0.45}$ | $79.27_{\pm 0.20}$ | $75.01_{\pm 0.54}$ | $74.95_{\pm 0.61}$ | $73.77_{\pm 0.83}$ | $74.39_{\pm 0.71}$ |
| OGBN-Arxiv | $55.50_{\pm 0.23}$ | $71.74_{\pm 0.29}$ | $73.65_{\pm 0.11}$ | $58.58_{\pm 0.11}$ | $58.68_{\pm 0.17}$ | $57.79_{\pm 0.56}$ | $58.62_{\pm 0.05}$ |
| OGBN-Products | $61.06_{\pm 0.08}$ | $75.79_{\pm 0.12}$ | $79.45_{\pm 0.59}$ | $61.19_{\pm 0.23}$ | $61.31_{\pm 0.20}$ | $60.28_{\pm 0.80}$ | $61.60_{\pm 0.10}$ |
| FullCora | $33.54_{\pm 0.64}$ | $61.06_{\pm 0.24}$ | $58.95_{\pm 0.55}$ | $57.13_{\pm 0.37}$ | $57.25_{\pm 0.43}$ | $56.29_{\pm 0.17}$ | $56.73_{\pm 0.41}$ |

## E ADDITIONAL EXPERIMENTAL RESULTS

### E.1 ABLATION STUDY ON MORE GRAPH OPERATORS

In this section, we further validate the effectiveness of GraphAny by gradually adding diverse types of graph convolution for propagating features. Specifically, we consider three types of graph convolutions:

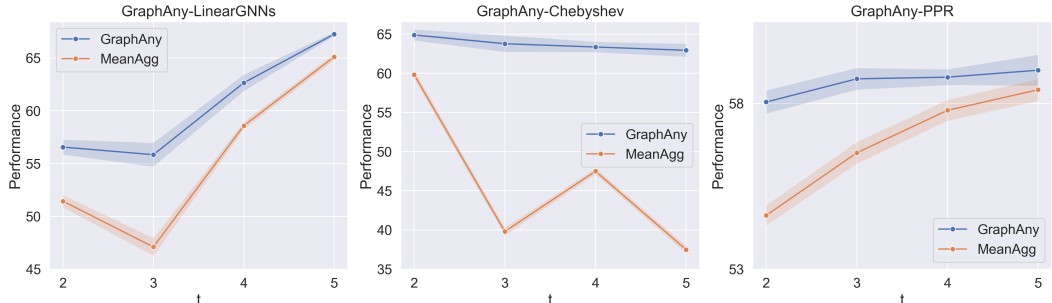

Figure 10: Ablation study with different graph operators (test accuracy in %): GraphAny-LinearGNN (left), GraphAny-Chebyshev (middle), and GraphAny-PPR (right). GraphAny is trained on the Wisconsin dataset and evaluated on all 31 datasets.

- **GraphAny-LinearGNN** gradually adds the following graph convolution operators: $\boldsymbol{F} = \boldsymbol{X}$ (Linear), $\boldsymbol{F} = (\boldsymbol{I} - \bar{\boldsymbol{A}})\boldsymbol{X}$ (LinearHGC1), $\boldsymbol{F} = (\boldsymbol{I} - \bar{\boldsymbol{A}})^2\boldsymbol{X}$ (LinearHGC2), $\boldsymbol{F} = \bar{\boldsymbol{A}}\boldsymbol{X}$ (LinearSGC1), and $\boldsymbol{F} = \bar{\boldsymbol{A}}^2\boldsymbol{X}$ (LinearSGC2), with $\bar{\boldsymbol{A}}$ denoting the row normalized adjaceny matrix. In this case, when increasing $t$ high-pass and low-pass predictions are gradually added.

- **GraphAny-Chebyshev** gradually adds the Chebyshev filters (Defferrard et al., 2016), where $\boldsymbol{F}^{(t)}$ is computed recursively by:

$$
\begin{aligned}
\boldsymbol{F}^{(1)} &= \boldsymbol{X}, \\
\boldsymbol{F}^{(2)} &= \hat{\boldsymbol{L}}\boldsymbol{X}, \\
\boldsymbol{F}^{(t)} &= 2\hat{\boldsymbol{L}}\boldsymbol{F}^{(t-1)} - \boldsymbol{F}^{(t-2)},
\end{aligned}
\tag{16}
$$

and $\hat{\boldsymbol{L}}$ denotes the scaled and normalized Laplacian $\frac{2\boldsymbol{L}}{\lambda_{\max}} - \boldsymbol{I}$ with $\boldsymbol{L} = \boldsymbol{I} - \boldsymbol{D}^{-1/2}\boldsymbol{A}\boldsymbol{D}^{-1/2}$. In this case, when increasing $t$ high order predictions are gradually added.

- **GraphAny-PPR** gradually adds the personalized PageRank with increasing restart ratios: $\{0.01, 0.25, 0.5, 0.75, 0.99\}$. In this case, when increasing $t$ local predictions are gradually added.

We compare the performance of GraphAny against the baseline, which averages all channels, denoted as *MeanAgg*. The results are shown in Figure 10, where we have the following observations:

**GraphAny consistently outperforms the baseline MeanAgg by a significant margin**, demonstrating effective and consistent inductive knowledge transfer across multiple datasets with different types of graph convolutions. **The performance varies across graph convolution types.** For LinearGNNs, low-pass graph convolutions (LinearSGC1 and LinearSGC2) significantly enhance performance. In contrast, for Chebyshev graph convolutions, adding high-order convolutions reduces performance as higher-order information introduces more noise. For personalized PageRank, adding more local channels consistently improves results.

### E.2 COMPARISON AGAINST ACM-GNN

To comprehensively compare GraphAny's performance, we included ACM-GNN (Luan et al., 2022) as a baseline due to its good performance on both homophilic and heterophilic graphs. However, ACM-GNN faces scalability issues that prevent its application to large graphs. Using the authors' implementation[2], we encountered out-of-memory for a GPU of 40GB on four large datasets: Questions, Reddit, Arxiv, and Product. Therefore, we evaluated ACM-GNN on the remaining 27 graphs, with results reported in Table 6.

Our observations are as follows: Tuning ACM-GNN is highly time-consuming, requiring 672 GPU hours on 27 graphs, while GraphAny requires only 4 GPU hours, showcasing its efficiency and the advantage of inductive inference. In terms of performance, GraphAny outperforms ACM-SGC

---

[2]https://github.com/SitaoLuan/ACM-GNN/tree/main/ACM-Pytorch/

and is only slightly (1-2%) below ACM-GCN. However, this slight difference is not a significant disadvantage for GraphAny, given the unfair advantage transductive models have by leveraging the inductive validation set for parameter and hyperparameter tuning, as well as the substantial difference in runtime.

Table 6: Experimental results of ACM-GNNs. Hyperparameter search durations: ACM-GNNs (672 GPU hours), MLP/GCN/GAT (240 GPU hours total), GraphAny (denoted as Ours, 4 GPU hours).

| Metric | MLP | GCN | GAT | ACM-SGC | ACM-GCN | Ours-Cora | Ours-Wis | Ours-Arxiv | Ours-Product |
|---|---|---|---|---|---|---|---|---|---|
| Heldout (25 graphs) | 54.87 | 64.19 | 64.01 | 60.91 | 66.92 | 65.11 | 65.23 | 65.56 | 65.56 |
| Overall (27 graphs) | 55.07 | 63.83 | 64.26 | 61.82 | 67.79 | 65.58 | 66.02 | 66.12 | 66.14 |

