# OpenReview forum: "Fully-inductive Node Classification on Arbitrary Graphs"
_ICLR.cc/2025/Conference — ICLR 2025 Poster_

### Official Review · Reviewer_SumM · 2024-10-24

**Soundness:** 3
**Presentation:** 3
**Contribution:** 4
**Rating:** 6
**Confidence:** 3

**Summary:**

The paper introduces GraphAny, a model designed for fully-inductive graph learning, where models must infer on new graphs with varying structures, features, and labels. GraphAny leverages LinearGNN for analytical graph inference, adaptable to diverse graph types. By integrating multiple LinearGNN predictions using learned inductive attention, GraphAny ensures robust generalization to new graphs. Empirical results demonstrate GraphAny's effectiveness, achieving a 67.26% average accuracy on 30 new graphs with minimal training data, outperforming both inductive and transductive baselines.

**Strengths:**

1. this paper raises a more general and challenging task for graph ood, that is the fully inductive node classification, which requires the model can generalize to arbitrary graphs, involving new structures, new dimensions, and semantics for their feature and label spaces.

2. this paper designs a novel method GraphAny, that integrates the multiple LinearGNN predictions and learned inductive attention, which satisfies the permutation invariant and robust to dimension changes

3.  the paper gives comprehensive experiments and evaluation of various datasets, showing the effectiveness of their methods.

**Weaknesses:**

1. The lack of baseline. This paper seems only to compare with the test-adapted GNN models as the baselines (GCN, GAT, MLP), I am not very certain if any other GNN baselines trained on the one dataset while generalizing to more datasets, such as the LLM-based GFM[1].

2. Since your method is based on the combination of 5 different linearGNNs ($ F = X$ (Linear), $F = AX$ (LinearSGC1), $F = A^2X $(LinearSGC2), $F = (I − A)X$ (LinearHGC1) and $F = (I − A)^2X$ (LinearHGC2) ), have you ever compared your method with the random coefficients combination of them? I suggest comparing GraphAny to a baseline that uses random or fixed coefficients to combine the 5 LinearGNN components. This would help isolate the benefit of the learned inductive attention mechanism.

[1] One for All: Towards Training One Graph Model for All Classification Tasks

**Questions:**

1.  Could you compare your method with the random coefficients combination of 5 different linearGNNs?

2. According to your Table 2 and Figure 7, it seems that SGC1 and SGC2 occupy a dominant position( high weight and high accuracy). Could you discuss why this happens more? Could you analyze why SGC1 and SGC2 tend to get higher weights and accuracy? Does this suggest that simpler graph convolutions are more transferable? How might this insight inform future designs of inductive graph models?

---

> ### Author Response · Authors · 2024-11-22
> **Rebuttal by Authors**
>
> We thank the reviewer for acknowledging the merits of our work. Below, we would like to comment on the identified weaknesses and answer your questions.
>
> **W1. Lack of baselines that generalize to more datasets.**
>
> We agree with the reviewer that some LLM-based approaches can generalize to new datasets based on text descriptions. However, this paper aims to study a more fundamental setting of generalizing across datasets purely based on arbitrary continuous features and categorical labels. LLM-based approaches can’t fit into this challenging setting, since most datasets don’t have text descriptions in our experiments.
>
> While GCN and GAT seem to be not strong enough, we additionally consider stronger baselines that work well for both homophilic and heterophilic datasets, including ACM-SGC and ACM-GNN [Luan et al., 2022]. As shown in Table 6, GraphAny achieves better or comparable performance using much less computational resources (a detailed discussion is provided in Appendix E.2).
>
>
> **W2 and Q1: Comparison of GraphAny against a fixed combination of 5 LinearGNN components.**
>
> We thank the reviewer for the feedback. We have those results in Figure 9 actually (although not explicitly discussed). The transductive results at batch 0 (green line) are the requested random-initialization result, which are much worse than the final performance of GraphAny.
>
> Besides, we provide additional results in Figure 10, where GraphAny is compared with the baseline that averages all the predictions (denoted as MeanAgg). It is obvious that GraphAny consistently outperforms the baseline MeanAgg by a significant margin, demonstrating the necessity of learning the coefficients rather than using fixed ones.
>
>
> **Q2: Any insight why LinearSGC1 and LinearSGC2 tend to get higher weights.**
>
> Your observation that LinearSGC1 and LinearSGC2 are the most effective graph convolution kernels is correct. This finding aligns with the standard homophily assumption, which suggests that connected nodes are likely to share similar labels, making simpler convolutional kernels like LinearSGC1 and LinearSGC2 particularly effective in such settings.
>
> Regarding transferability, we believe that simpler graph kernels might exhibit stronger inductive generalization due to introducing less inductive bias. This reduced bias allows these kernels to generalize better to new graphs. However, it is important to note that these simple kernels might also have less expressive power and, hence, weaker transductive performance, as they may not fully fit the complex distributions during training. Therefore, we believe that there exists a potential tradeoff for designing inductive graph models: one might need to balance transductive performance (specialized for a specific graph) and inductive generalization (transferability to new graphs).
>
> **Reference**
>
> [Luan et al., 2022] Revisiting Heterophily For Graph Neural Networks. NeuriPS 2022.

---

> > ### Comment · Reviewer_SumM · 2024-11-22
> >
> > Thank you very much for your experiments and the explanation, which solves my concerns.
> >
> > I have two more questions about ACM-GNN.
> >
> > - You said in Appendix E.2 that tuning ACM-GNN requires 672GPU while GraphAny requires 4GPU hours. Why does ACM-GNN consume a lot of time? I understand that ACM-GNN is a quite complex model, is that the reason?
> >
> > - Also I know that ACM-GNN is not for Arbitrary Graphs, it is for heterophilic graphs. There is one thing I am more curious about, Are there any baselines specialized for Arbitrary Graphs?
> >
> > Thank you again for your reply.

---

> > > ### Author Response · Authors · 2024-11-22
> > > **Response to Rebuttal**
> > >
> > > Thank you for your quick reply. We're glad to know that our responses have addressed some of your concerns.
> > >
> > > **Q1. Why is GraphAny significantly faster than ACM-GNN?**
> > >
> > > GraphAny's speed advantage over ACM-GNN can be attributed to several factors. Firstly, ACM-GNN is a transductive model that requires end-to-end training on each graph, resulting in the training of 31 models, each with its own set of parameters and hyperparameters. In contrast, GraphAny trains on a single graph with one consistent set of parameters/hyperparameters. Additionally, GraphAny enjoys better time complexity; its preprocessing occurs in $O(|\mathcal{E}|)$ time, and each epoch processes in $O(|\mathcal{V}_L|)$ time, as opposed to $O(|\mathcal{E}|)$ for ACM-GNN.
> > >
> > > **Q2. Are there any baselines specialized for Arbitrary Graphs?**
> > >
> > > To the best of our knowledge, we made the first attempt to address the fully-inductive learning problem on arbitrary graphs. Currently, GraphAny is the only model that supports inductive learning across arbitrary graphs.

---

> > > > ### Comment · Reviewer_SumM · 2024-11-23
> > > >
> > > > Thank you for your reply which explains my concerns. I think this paper is a really aggressive attempt in the graph field. I will keep my positive score.

---

> > > > > ### Author Response · Authors · 2024-11-24
> > > > > **Response to Reviewer**
> > > > >
> > > > > Thank you for your thoughtful feedback.
> > > > > We are glad our responses addressed your concerns.

---

### Official Review · Reviewer_GtqS · 2024-11-01

**Soundness:** 3
**Presentation:** 2
**Contribution:** 4
**Rating:** 8
**Confidence:** 4

**Summary:**

The authors focus on addressing a challenging problem: enabling GNNs to be fully-inductive across diverse datasets. They propose a model called GraphAny. Specifically, the authors employed multiple encoders (LinearGNNs) whose parameters can be obtained analytically, allowing it to generalize across datasets with different feature and label spaces. Additionally, the authors design an attention-based, learnable MLP to capture transferable graph patterns. Extensive experiments demonstrate the model's effectiveness.

**Strengths:**

S1. The authors are ambitious, tackling a highly challenging and valuable problem: designing a foundational GNN model that can generalize across diverse datasets.

S2. The proposed method is ingenious. The authors introduce a LinearGNN that does not require training, enabling the model to adapt to different datasets.

S3. The experimental results are powerful and impressive.

S4. The authors provide the complete code, along with a well-organized README file, to support their views.

**Weaknesses:**

W1. In fact, the proposed LinearGNNs seem to me more like a data preprocessing method which requires no learning to unify the feature and label spaces through analytical solutions.

W2. Regarding W1, the authors’ statement in the Introduction that GraphAny is the first fully-inductive method seems somewhat over-claimed. According to the views in this paper, any model that can be solved analytically (i.e., without training) could also be seem as fully-inductive. Nonetheless, this point does not negate the contribution of the attention-based component to knowledge transfer.

W3. The paper does not mention some recent methods capable of achieving the fully-inductive as described, such as GraphControl [1].

W4. I suggest that the author should provide the data split ratio for downstream test datasets (it is vaguely mentioned only in the appendix). This is a crucial setting, as if my understanding is correct, the proposed method requires a certain amount of ground-truth labels to analytically solve the parameters of LinearGNNs on test datasets.

W5. Based on W4, the approach in this paper seems to be semi-supervised (or fine-tuned) on downstream tasks, meaning it has access to the same amount of labeled data as other semi-supervised algorithms like GCN. Moreover, GraphAny benefits from additional prior knowledge from other datasets (i.e., the pre-training phase), making it seemingly more advantageous compared to other algorithms in experimental settings. This stands in contrast to the authors' claim that other semi-supervised algorithms have an additional advantage over GraphAny in the experimental settings.

If LinearGNNs do not require any test dataset labels to solve the parameters (i.e. completely zero-shot scenario), then W4 and W5 would not hold. I strongly recommend that the authors add further explanations in the paper to improve reader comprehension.

[1] GraphControl: Adding Conditional Control to Universal Graph Pre-trained Models for Graph Domain Transfer Learning, WWW24.

**Questions:**

Please see above.

---

> ### Author Response · Authors · 2024-11-22
> **Rebuttal by Authors**
>
> We thank the reviewer for acknowledging the merits of our work. Below, we would like to comment on the identified weaknesses.
>
> **W1. LinearGNN is a data preprocessing method.**
>
> Yes, your understanding is correct. We recognize LinearGNN as a fast and solvable solution for node classification, which enables GraphAny’s inductive inference without training. We don’t think this is a weakness, since GraphAny learns inductive attention scores to combine predictions from multiple LinearGNNs.
>
> **W2. Any model that can be solved analytically is fully-inductive. GraphAny is overclaimed.**
>
> Fully-inductive not only implies inference on any graph, but also **generalization** to new graphs. We would like to highlight that LinearGNNs, and other GNNs with solvable weights actually yield a transductive function $f=\mathbb{R}^{d} \rightarrow \mathbb{R}^c$ and cannot generalize to any new datasets with different feature and label spaces that require $f’=\mathbb{R}^{d’} \rightarrow \mathbb{R}^{c’}$.
>
> By contrast, the function $f_\theta: \mathbb{R}^{t(t-1)} \rightarrow \mathbb{R}^t$ learned by GraphAny operates on a fixed dimension input regardless of the dataset, and thereby generalizes to new datasets. Hence, GraphAny should be considered as fully-inductive.
>
> **W3. A recent work, GraphControl, is also fully-inductive.**
>
> Thank you for bringing the recent related work on GraphControl to our attention. We have reviewed this work and acknowledge its relevance. However, GraphControl is still a transductive model with the pre-training-fine-tuning pipeline, which cannot transfer to new graphs with different label spaces. For example, the graph-prompt features and the output layer for prediction must be learned end-to-end for a new graph task. In contrast,  we focus on the more challenging fully-inductive setting, where the model must generalize to unseen graphs *without additional training*. Therefore, the presence of this related work does not diminish the unique contribution of GraphAny. We’ve also updated the manuscript and discussed GraphControl in the related work session.
>
> **W4. Provide the data split ratios for test datasets.**
>
> We thank the reviewer for the comment and updated the dataset split ratio in Table 3. It’s noteworthy to highlight that we did not tweak the dataset training ratio to make our results look better. We strictly follow the splits of the original data source (e.g. DGL and PyG) if they exist. For those that don’t provide splits, we follow the standard semi-supervised settings (20 labeled nodes per class for training and the same amount of data for valid and test sets).
>
> **W5. GraphAny has an advantage over the setting of semi-supervised learning, not in the opposite way.**
>
> We respectfully disagree with your claim that GraphAny has an additional advantage over the semi-supervised training baselines. The ways where semi-supervised models and GraphAny use training labels are totally different: semi-supervised models use training labels to perform gradient descent, while GraphAny only uses training labels for inference, without changing any of its parameters. We emphasize that semi-supervised models have to train 31 separate models (with 31 different sets of hyperparameters) in order to perform inference on 31 datasets. By contrast, GraphAny only trains 1 model using 1 dataset and can perform inference on 31 datasets. Since semi-supervised models additionally leverage the training labels to tune parameters and the validation sets to search hyperparameters, they are supposed to have an advantage over GraphAny.
>
> Besides, your understanding of LinearGNNs is correct, they do rely on the training labels of the inductive datasets to perform training-free inference.

---

> > ### Author Response · Authors · 2024-11-24
> > **Official Comment by Authors**
> >
> > Dear Reviewer GtqS,
> >
> > Thank you for your valuable comments and suggestions to improve our paper. In response to your feedback, we have clarified the differences between the proposed fully-inductive setting, analytical solutions, and transductive settings. Additionally, we have updated the manuscript to include a discussion on GraphControl and provided detailed explanations of the data split ratios.
> >
> > We would appreciate it if you could confirm whether our responses have adequately addressed your concerns.
> >
> > We look forward to your feedback and thank you for your time and consideration.

---

> > > ### Comment · Reviewer_GtqS · 2024-11-25
> > >
> > > Thanks for your response. I have carefully read the authors reply and other reviews. Besides, I have checked the code, as I am very interested in this paper. In my opinion, for each dataset, GraphAny needs to retrain multiple LinearGNNs using method like least squares, which makes this part non-transferable. However, the authors innovatively proposed the Inductive Attention component, which serves as a Fully-Inductive result selector. My main concerns has been addressed. Therefore, I have decided to raise my score from 6 to 8.
> > >
> > > Well, I still cannot fully agree with your claim that GraphAny is the first fully-inductive method. This is because GraphAny still requires some labeled data for downstream training (which is crucial for the quality of LinearGNNs as they are non-transferable). As I mentioned earlier, methods like GraphControl with appropriate prompts can be applied in few-shot scenarios (e.g., 3-shot or 5-shot) or even zero-shot, whereas GraphAny requires 20-shot or more. From the perspective of required labels for downstream tasks, GraphAny does not hold a clear advantage. Nonetheless, this does not diminish my recognition of the paper's significant innovation and valuable insights.

---

> > > > ### Author Response · Authors · 2024-11-25
> > > > **Response to Reviewer Feedback**
> > > >
> > > > Thank you for your thoughtful and detailed feedback, as well as the time and effort you have dedicated to reviewing our work. We greatly appreciate your interest in GraphAny and the insights you have shared.
> > > >
> > > > Your understanding of how GraphAny is trained is indeed precise. To our understanding, the graph-specific (transductive) part that leverages labeled nodes to predict is essential due to the varying input and output spaces across graphs. However, GraphAny demonstrates that a transferable function—namely, the proposed inductive attention—can be successfully applied across graphs, which we view as a key contribution.
> > > >
> > > > We would also like to clarify a point regarding GraphControl. While GraphControl can operate in few-shot scenarios, it cannot perform true zero-shot inference. As outlined in Algorithm 1 of the GraphControl paper, fine-tuning is still required for new datasets to learn the MLPs and output layer specific to each graph. Nonetheless, we acknowledge and appreciate your suggestion, and we have incorporated it to further enrich the related work section of our paper.
> > > >
> > > > Thank you once again for your valuable feedback and for raising your score based on the addressed concerns. Your recognition of GraphAny’s innovations means a lot to us.

---

### Official Review · Reviewer_o9Mt · 2024-11-03

**Soundness:** 3
**Presentation:** 2
**Contribution:** 2
**Rating:** 6
**Confidence:** 4

**Summary:**

This paper tackles the issue of fully inductive graph learning by introducing GraphAny. The proposed method consists of LinearGNN to preprocess the features following the idea of SGC and attention module to transform the feature.

**Strengths:**

1. This paper tackles a great challenging fully-inductive graph learning task.
2. This paper introduces an inductive attention module that satisfies permutation invariance properties and generalizes to new graphs.

**Weaknesses:**

1. The presentation of this paper needs improvement. Many details are missing in the section of methodology.
- The authors conduct the experimental on motivating the entropy normalization, while the experimental setup in figure 5 is not explicit. It's not suggested to specify what these methods are until the section 4.1. The authors should provide more explicit explanation of the experimental setup for Figure 5.
- It's not clear what is the learnable parameters in the attention module and how to get the attention vector $\alpha$. A clear description of the learnable parameters in the attention module should be added.
- It's weird to call $y_u^{(i)}$ in equations 9 and 10 as node feature and it's more proper to describe it as label vector considering its dimensionality.

2. In figure 3, the authors mention that LinearGNN is non-parametric, but LinearGNN involves the learnable weight matrix W in equation 1. It's improper to claim that LInearGNN is a non-parametric solution. The authors should revise their description of LinearGNN to avoid confusion.

3. This paper mentions that it is always possible to cheat the fully-inductive setup by training a separate instance of existing models for each test dataset (in Line 75). However, the proposed LinearGNN operates like what it just said by training a linear layer with a graph convolution operation for a test graph and the authors called this LinearGNN a non-parametric solution, or preprocessing step (in Table 1). It's hard to convince the readers that the proposed method is a fully-inductive graph learning method.

4. Though the authors show that GraphAny has better average performance in total 31 graphs in Table 2. However, the experimental results in Table 5 shows that GAT outperforms GraphAny in 18 out of 31 graphs, which means that the proposed method does not have advantage in the fully inductive learning setting. In addition, GAT is a baseline proposed in 2018, and many recent methods can outperform GAT in most of these graphs.

5. How does the different values of t influence the performance of GraphAny on different datasets? It's better to include an ablation study on the effect of t.

**Questions:**

1. How do you get the attention score in equation 5? Do you just sum all elements in matrix $P_u^{i}$ in equation 10? What is the learnable weight in the attention module as shown in figure 3?

2. Can you further explain the experimental setting in figure 5? What does the density mean? Since the value is in the range of [0, 1], is this value normalized?

3. This paper mentions that it is always possible to cheat the fully-inductive setup by training a separate instance of existing models for each test dataset (in Line 75). However, the proposed LinearGNN operates like what it just said by training a linear layer with a graph convolution operation for a test graph and the authors called this LinearGNN a non-parametric solution, or preprocessing step (in Table 1). It's hard to convince the readers that the proposed method is a fully-inductive graph learning method. Can the authors clearly differentiate your approach from the "cheating" setup?

---

> ### Author Response · Authors · 2024-11-22
> **Rebuttal by Authors**
>
> We sincerely thank the reviewer for pointing out areas of improvement for our paper. Below, we address each of the identified weaknesses:
>
> **W1. Many methodology details are missing**
>
> W1.1 We sincerely thank the reviewer for the constructive feedback on the experimental settings of Figure 5. We’ve updated the caption of Figure 5 to make it clear and self-contained.
>
> W1.2 The attention vector $\alpha$ is predicted by the attention module $f_\theta$ based on the entropy-normalized features, as shown in the bottom right of Figure 3. In practice, we implement $f_\theta$ as an MLP and all the learnable parameters are the parameters of that MLP, as stated in line 330-334.
>
> W1.3 We thank the reviewer for pointing this out. We call $\hat{y}_u^{(i)}$ node features because they serve as input to our inductive attention module for node $u$. We understand this slightly abuses the term “node features” used in node classification. To distinguish  $\hat{y}_u^{(i)}$ against $X$, we have revised to call $\hat{y}_u^{(i)}$ “features for inductive attention”.
>
> **W2: LinearGNNs are not non-parametric since they involve a learnable weight matrix W.**
>
> By non-parametric, we refer to having no *learnable* parameters.  Although LinearGNNs do have weights (which we solve analytically through the pseudoinverse), they are not learnable. Besides, it is noteworthy to point out that the LinearGNNs are still transductive models that do not transfer any knowledge across graphs, whereas GraphAny learns inductive functions of LinearGNNs predictions that can generalize to unseen graphs without additional training.
>
> **W3 and Q3: Why are transductive models cheating in the fully-inductive setting?**
>
> For the standard transductive setting, separate models with **different sets of parameters** and hyperparameters are trained for different datasets. In fully-inductive settings, **one set of parameters** is trained for different datasets and generalize to new graph **without additional training**. That said, transductive models cannot perform fully-inductive inference in the first place as the input and output spaces vary. However, the transductive models can be viewed as strong baselines for inductive models: as they leverage unobserved validation sets of inductive data to optimize parameters when training the model (e.g. optimizer, learning rate, dropout ratio, number of epochs) and tuning hyperparameters (e.g. selecting the depth of graph convolutions).
>
> As mentioned in Section 3.3 and our responses in W1.2, GraphAny is a fully-inductive model, which does not model the input and output spaces and learns to fuse different predictions. Once trained, the learned inductive attention $f_\theta: \mathbb{R}^{t(t-1)} \rightarrow \mathbb{R}^t$ is ready to generalize to arbitrary graphs.
>
>
> **W4. GAT outperforms GraphAny on 18 out of 31 datasets.**
> First, as we have mentioned in our response to W3, it is unfair to compare a fully-inductive model to a transductive model like GAT as transductive models are trained on known labeled nodes and leverage validation sets to select hyperparameters for each of the 31 datasets. GraphAny runs inference on all new unseen graphs outside its small training set (eg, training on Wisconsin and running inference on 30 other graphs). Nevertheless, even in the fully-inductive setting, GraphAny outperforms several transductive baselines.
>
> To further ease your concerns, we added two recent baselines, ACM-SGC and ACM-GNN [Luan et al., 2022], which have good homophilic and heterophilic performance. As shown in Table 6, GraphAny achieves better or comparable performance using much less computational resources (a detailed discussion is provided in Appendix E.2).
>
> **W5. How does the different values of t influence the performance of GraphAny?**
> We thank the reviewer’s advice and have added the ablations on t in Appendix E.1. As shown in Figure 10, for different graph convolution operators, such as LinearGNNs, Chebyshev polynomials, and personalized page rank, increasing t has diverse effects. For LinearGNNs, increasing t and adding low-pass graph convolutions (LinearSGC1 and LinearSGC2) significantly enhance performance. In contrast, for Chebyshev graph convolutions, adding high-order convolutions reduces performance. For personalized PageRank, adding more local channels consistently improves results.

---

> ### Author Response · Authors · 2024-11-22
> **Rebuttal by Authors (cont.)**
>
> Here, we discuss the proposed questions:
>
> **Q1. How to get the attention scores?**
>
> We don’t sum up the $P_u^{t}$ in the Equation 10 to obtain the attention score. The $P_u(i,j)$ is  a dimension of the entropy-normalized feature for the attention function. As explained in Section 3.3, The learnable weights of GraphAny is the inductive attention module$f_\theta: \mathbb{R}^{t(t-1)} \rightarrow \mathbb{R}^t$, which computes the attention score for fusing LinearGNN predictions based on their interactions.
>
> **Q2. Experimental settings in Figure 5.**
>
> The experiments in Figure 5 illustrate the probability distribution of different features for inductive attention. Specifically, we compare the distributions between Euclidean distances (the first row) and entropy-normalized (the second row) features between five channels: $\boldsymbol{F} = \boldsymbol{X}$ (Linear), $\boldsymbol{F} = \bar{\boldsymbol{A}} \boldsymbol{X}$ (LinearSGC1), $\boldsymbol{F}=\bar{\boldsymbol{A}}^2 \boldsymbol{X}$ (LinearSGC2), $\boldsymbol{F}=(\\boldsymbol{I}-\bar{\boldsymbol{A}}) \boldsymbol{X}$ (LinearHGC1) and $\boldsymbol{F} = (\boldsymbol{I} - \bar{\boldsymbol{A}})^2 \boldsymbol{X}$ (LinearHGC2) with $\bar{\boldsymbol{A}}$ denoting the row normalized adjaceny matrix. The density means the estimated density of the empirical distribution function based on the observed distribution of distances/features. The Euclidean distances are computed by features normalized to unit length, the entropy-normed features are normalized via dynamically determining the sigma for each node (check the explanation under equation 10).
>
> **Q3. Why transductive models are cheating?**
>
> Please refer to our response to W3.
>
>
>
> **Reference**
>
> [Luan et al., 2023] Revisiting Heterophily For Graph Neural Networks. NeurIPS 2023.

---

> > ### Author Response · Authors · 2024-11-24
> > **Official Comment by Authors**
> >
> > Dear reviewer o9Mt,
> >
> > Thank you for your valuable comments and suggestions for improving our paper. Following your feedback, we have explained the methodology details and clarified our statement regarding transductive models and LinearGNN. We have also updated our manuscript to make it more comprehendible. We would like to confirm whether our response has adequately addressed your concerns.
> >
> > We look forward to your feedback.

---

> > > ### Comment · Reviewer_o9Mt · 2024-11-24
> > > **Reply to Authors' Rebuttal**
> > >
> > > Thank you for the detailed explanation and the additional experimental results. You have addressed my concerns. I will increase my score to 6.

---

> > > > ### Author Response · Authors · 2024-11-25
> > > > **Comment by Authors**
> > > >
> > > > Thank you for your response and for updating your score!
> > > > We are glad our responses addressed your concerns.

---

### Official Review · Reviewer_eX9V · 2024-11-04

**Soundness:** 3
**Presentation:** 2
**Contribution:** 4
**Rating:** 8
**Confidence:** 4

**Summary:**

This paper studies the problem of fully inductive node classification, where limited parameters are learned from one small graph, and inference other unseen graphs.  The authors propose GraphAny, which consists of two components, one is a set of linearGNNs, and the other is a learnable attention MLP function. Using pseudo-inverse, LinearGNNs directly compute the node embeddings of corresponding linearGNN channels. Then a sophisticated attention technique which has properties of permutation-invariant and robust dimension generalization is used to combine these embeddings. The extensive experiments show that GraphAny gains significant improvements over the state-of-the-art methods in many datasets.

**Strengths:**

1. This paper proposes a novel problem setting seemingly impractical, and provides a reasonable solution to it. Previously, I was doubtful about the feasibility of graph foundation models, since unlike in NLP and CV, graph data is more universal and diverse. The information heterogeneity between different graphs may make this fully inductive setting impossible, i.e., I didn't think the knowledge in different graphs has much in common. However, the authors provide an impressive and valid solution to this problem, which is a good contribution to the community.

2. The proposed method is well-motivated and well-designed. The attention module that tackles dimensionality and permutation issues is particularly novel and interesting, with strong intuition.

3. The experiments are extensive and convincing. An impressive number (31) of datasets are involved in this fully-inductive setting, and the good average score of GraphAny demonstrates its effectiveness.

4. The ablation study is comprehensive and insightful. The authors provide a clear understanding of the importance of each component in GraphAny.

**Weaknesses:**

1. (Explainability) I didn't see any explanation of one very important question: why could the knowledge learned from one graph be transferred to another unseen and unrelated graph? The authors should provide more intuitive insights on this point. From my point of view, LinearGNNs with different graph operations may serve as probes to extract different types of intrinsic knowledge from the graph, then the permutation and dimension invariant attention module could combine this knowledge in a semantic space where the common knowledge of graphs is shared. The authors should provide more insights on this point, i.e., why it works well.

2. (Experiments) Although the proposed AnyGraph shows a high average performance, it is not the best in all datasets, especially in some large datasets such as Arxiv, Reddit and Products. I don't think homophily could explain this, since AnyGraph (Arxiv) also performs poorly. The authors could provide more insights on why AnyGraph fails in these datasets, and how to possibly improve it.

3. (Experiments) The transductive baselines (GCN, GAT) are not strong enough. Since the benchmark contains so many datasets ranging from highly homophily to highly heterophily, baselines [1,2,3] that could fit both homophilous and heterophilous graphs should be compared. I highly recommend the authors to add some of these baselines to make the results more convincing.


[1] Luan, S., Hua, C., Lu, Q., Zhu, J., Zhao, M., Zhang, S., ... & Precup, D. (2022). Revisiting heterophily for graph neural networks. Advances in neural information processing systems, 35, 1362-1375.

[2] Lim, D., Hohne, F., Li, X., Huang, S. L., Gupta, V., Bhalerao, O., & Lim, S. N. (2021). Large scale learning on non-homophilous graphs: New benchmarks and strong simple methods. Advances in Neural Information Processing Systems, 34, 20887-20902.

[3] Zhu, J., Rossi, R. A., Rao, A., Mai, T., Lipka, N., Ahmed, N. K., & Koutra, D. (2021, May). Graph neural networks with heterophily. In Proceedings of the AAAI conference on artificial intelligence (Vol. 35, No. 12, pp. 11168-11176).

**Questions:**

Most of the questions and suggestions are already mentioned in the weaknesses section. I would like to mention some minor points here.

1. I would like to see more graph operations used in LinearGNN instead of just X, AX, A^2X, (I-A)X, (I-A)^2X. For example, the Chebyshev polynomial operation, the PageRank operation, normalized Laplacian operation, etc. I think more operations could provide more diverse perspectives of the graph, and thus improve the performance of GraphAny at a little extra cost.

2. I doubt the time complexity in Table 1, since pseudo-inverse is used in LinearGNN, which is computationally expensive up to O(n^2d), could the authors explain this?

---

> ### Author Response · Authors · 2024-11-22
> **Rebuttal by Authors**
>
> We thank the reviewer for the constructive feedback. Below we would like to comment on the identified weaknesses.
>
> **W1. Why can GraphAny transfer to unseen graphs?**
>
> Above all, **the proposed distance features capture the node-level message-passing pattern** and infer the optimal attention for fusing LinearGNN predictions. Here we give an intuitive example: We know that LinearSGC2 is a good prediction channel for homophilic graphs and MLP is a good channel for heterophilic graphs (Figure 7 left). Hence, for an unseen graph, a logical guess would be to assign higher attention weights to LinearSGC2 if the graph displays homophilic properties. From Figure 5, we observe that homophilic and heterophilic information can be inferred from the entropy-normalized features (bottom row), where homophilic graphs like Cora, Arxiv, and FullCora exhibit similar patterns. These message-passing patterns are leveraged by GraphAny to predict inductive attention scores.
>
> Moreover, it is important to note that our **inductive attention operates at the node level**, as specified in Equation 5. This means that even given an unseen graph with new structure/feature/label spaces, as long as the node of interest exhibits distance feature patterns akin to those of a node in a previously observed graph, GraphAny will likely assign similar attention weights. Luckily, even when training on one small graph, say Wisconsin with 120 nodes, there might be sufficiently diverse node-level message passing patterns, as observed in [Luan et al., 2022]. This enables the remarkable transferability of GraphAny to transfer to unseen graphs even trained on one dataset.
>
>
> **W2. Insights on why GraphAny fails on some datasets and how to improve it.**
>
> The reason why GraphAny doesn’t outperform transductive baselines on some datasets is that GraphAny is evaluated in a more challenging inductive setting. Specifically, the transductive models have additional advantages compared with GraphAny by leveraging unobserved validation sets of inductive data to optimize parameters when training the model (e.g. optimizer, learning rate, dropout ratio, number of epochs) and tuning hyperparameters (e.g. selecting the depth of graph convolutions).
>
> Second, one limitation of GraphAny is that to achieve training-free inference, we leverage LinearGNNs with analytical solutions but with limited expressivity. As a consequence, the performance is bounded by the linear combination of those LinearGNNs. There are several straightforward solutions to improve the performance of GraphAny, though all at the cost of breaking the fully-inductive assumption. One possible solution is to relax this constraint and train transductive non-linear models first and learn how to combine their predictions using the interactions between them. Although sacrificing the training-free property of GraphAny, this should give a better transductive performance, especially for those datasets with complex non-linear patterns.
>
> **W3. Comparison against strong baselines that consider both homophily and heterophily.**
> Following the reviewer’s suggestion, we included the required ACM-GNN [Luan et al., 2022] as a baseline due to its good performance on both homophilic and heterophilic graphs. However, ACM-GNN faces scalability issues that prevent its application to large graphs. Using the [authors’ implementation](https://github.com/SitaoLuan/ACM-GNN/tree/main/ACM-Pytorch/), we encountered out-of-memory for a GPU of 40GB on four large datasets: Questions, Reddit, Arxiv, and Product. This prevents us from adding ACM-GNN to our main table. Hence, we evaluated ACM-GNN on the remaining 27 graphs, with results reported in Table 6.
>
> Our observations are as follows: Tuning ACM-GNN is highly time-consuming, requiring 672 GPU hours on 27 graphs, while GraphAny requires only 4 GPU hours (**168× more efficient**), showcasing its efficiency and the advantage of inductive inference. In terms of performance, GraphAny outperforms ACM-SGC and is only slightly (1-2\%) below ACM-GCN. However, this slight difference is not a significant disadvantage for GraphAny, given the unfair advantage transductive models have by leveraging the inductive validation sets for parameter and hyperparameter tuning, as well as the substantial difference in runtime.

---

### Meta-Review · Area_Chair_Yhk1 · 2024-12-11

**Metareview:**

This paper proposes a general graph neural network that can be applied to new graphs which may have different feature and label space. Authors show that the approach outperforms a wide range of methods. While some reviewers have concerns on the selection of baseline methods and a few experimental setups, authors were able to clarify in the rebuttal. Overall, reviewers agree that the paper presents an important algorithm and is technically sound.

**Additional Comments On Reviewer Discussion:**

Some reviewers had questions on the selection of baseline methods and clarity of experiment setup, which were addressed during rebuttal. A the end, all reviewers are fairly positive and thus AC-reviewer discussion was not needed.

---

### Decision · Program_Chairs · 2025-01-22

Accept (Poster)